# Tree canopy and snow depth relationships at fine scales with terrestrial laser scanning

Ahmad Hojatimalekshah[1], Zach Uhlmann[1], Nancy F. Glenn[1], Christopher A. Hiemstra[2], Christopher J. Tennant[3], Jake D. Graham[1], Lucas Spaete[4], Art Gelvin[2], Hans-Peter Marshall[1], James McNamara[1], Josh Enterkine[1]

[1]Department of Geosciences, Boise State University, Boise, 83725, USA
[2]US Department of Agriculture, Forest Service, Geospatial Management Office, Salt Lake City, 84138, USA
[3]US Army Corps of Engineers, Sacramento, CA, 95814, USA
[4]Minnesota Department of Natural Resources, Division of Forestry, Resource Assessment, Grand Rapids, 55744, USA

*Correspondence to*: Nancy F. Glenn (nancyglenn@boisestate.edu)

**Abstract.** Understanding the impact of tree structure on snow depth and extent is important in order to make predictions of snow amounts, and how changes in forest cover may affect future water resources. In this work, we investigate snow depth under tree canopies and in open areas to quantify the role of tree structure in controlling snow depth, as well as the controls from wind and topography. We use fine scale terrestrial laser scanning (TLS) data collected across Grand Mesa, Colorado, USA (winter 2016-2017), to measure the snow depth and extract horizontal and vertical tree descriptors (metrics) at six sites. We utilize these descriptors along with topographical metrics in multiple linear and decision tree regressions to investigate snow depth variations under the canopy and in open areas. Canopy, topography and snow interaction results indicate that vegetation structural metrics (specifically foliage height diversity (FHD)) along with local scale processes like wind and topography are highly influential on snow depth variation. Our study specifies that windward slopes show greater impact on snow accumulation than vegetation metrics. In addition, the results indicate that FHD can explain up to 27 % of sub-canopy snow depth variation at sites where the effect of topography and wind is negligible. Solar radiation and elevation are the dominant controls on snow depth in open areas. Fine scale analysis from TLS provides information on local scale controls, and provides an opportunity to be readily coupled with lidar or photogrammetry from uncrewed aerial systems (UAS), airborne, and spaceborne platforms to investigate larger-scale controls on snow depth.

## 1 Introduction

Forests are distributed across approximately half of the snow-covered landmasses on Earth during peak snow extent (Kim et al., 2017), with snow in nonpolar, cold climate zones accounting for 17 % of the total terrestrial water storage (Rutter et al., 2009; Guntner et al., 2007). Estimating the amount of water stored in this snowpack, the snow water equivalent (SWE), and its spatial distribution under various physiographic conditions, are crucial to providing water managers with parameters to accurately predict runoff timing, duration and amount, especially in a changing climate. Snowbound forested regions are

rapidly changing in forest cover composition (e.g. fire, insect outbreaks, thinning) (Nolin and Daly, 2006; Bewley et al., 2010; Gauthier et al., 2015). Understanding how forest characteristics affect snow distribution, as well as how we might model the relationships between forests and snow distribution will benefit water management objectives.

Generally, complex tree structure reduces snow deposition by increasing snow-canopy interception. However, canopy sheltering at windy sites can reverse the influence of interception on snow accumulation (Dickerson et al., 2017). Shading degrades incoming shortwave radiation while sheltering reduces the wind speed and turbulent heat transfer within canopies, resulting in longer snowmelt duration relative to open areas. The extinction of shortwave radiation by shading, however, can enhance sub-canopy longwave irradiance from tree trunks (Pomeroy et al., 2009). Reducing wind speeds within the canopy due to sheltering can also reduce the spatial heterogeneity of snow depth and extent (Qiu et al., 2011). Therefore, at windy sites studies have shown similar snow deposition in open areas as under the canopy (Dickerson et al., 2017), changing sub-canopy deposition and accumulation. These and other studies demonstrate the complexity between snow processes and vegetation in the presence of other predominant controls for modeling snow distribution and properties.

Forest canopy cover can be incorporated into watershed and regional scale models as subgrid parameterization via snow depletion curves by relating canopy cover distributions to fractional melt patterns (Dickerson-Lange, et al., 2015; Homan et al., 2011; Luce et al., 1999). Pixel-level binary or weighted snow depth correction factors in gridded models (Hedrick et al., 2018, Winstral et al., 2013) can be adjusted for canopy cover, as well as a hybridized approach that adjusts radiative inputs differently in open areas and forest gaps based on their size and relationship to the surrounding forest (Seyednasrollah and Kumar, 2014). In research pertaining to forest-snow processes, forest plots may be classified qualitatively (e.g., Dickerson-Lange et al., 2015; Pomeroy et al., 2009) or more recently, at larger scales, quantitatively with airborne lidar (e.g. Mazzotti et al., 2019). The use of lidar in spatially distributed modeling efforts is rapidly advancing (e.g. Hedrick et al., 2018, Painter et al., 2016) and understanding how best to describe forest characteristics (cover, structure, gaps, etc.) relevant to snow distribution is evolving (Jenicek et al, 2018, Mazzotti et al., 2019, Yang et al., 2020).

Airborne lidar has been used to describe snow depths in forests starting almost two decades ago (e.g. Hopkinson et al., 2004), and more recently, to describe the relationships among forest characteristics and snow distributions (e.g. Moeser et al., 2015a,b; Mazzotti et al., 2019; Zheng et al., 2016; Tennant et al., 2017). Realizing airborne lidar's capabilities to provide high-resolution snow depth and canopy measurements across large extents, studies have identified vegetation characteristics as drivers of snow depth variation. For example, canopy structure along with the forest canopy edge were driving factors that govern the snow depth distribution in a study of alpine climates (Mazzotti et al., 2019). Similarly, mean distance to canopy and canopy closure have been identified as strong metrics for predicting snow interception (Moeser et al., 2015a). In the wind dominant case, Trujillo et al. (2007) observed that snow depth variability occurs at larger scales than those related to vegetation. They found that when canopy interception is dominant and wind effect is minimal, the variation in snow depth is controlled by vegetation characteristics. Broxton et al. (2015) found canopy-snow interception and shading properties in transition zones result in different snow depths in comparison to the open and under-the-canopy regions. Further work in snow depth variability near forest edges acknowledges that snow depth variations are due to the effects of temperature, wind speed and direction, solar

radiation, and forest distribution (Currier and Lundquist, 2018). Recently, uncrewed aerial systems (UAS) have been utilized to measure snow depth using photogrammetric techniques (e.g. Structure from Motion (SfM)) in open and sparsely-forested areas (Buhler et al., 2016, Cimoli et al., 2017, Harder et al., 2016, Lee et al., 2021, Jacob et al., 2020). In addition, a lidar mounted on a UAS can be collected at different scans angles, making it a reliable source for sub-canopy measurements and across catchment scales (~ 5km$^2$) (Harder et al., 2020). UAS-based observations can fill measurement gaps in airborne and spaceborne lidar measurements, and provide the opportunity to assess forest and snow relationships at a spatial (and temporal) resolution higher than airborne lidar and from a different viewing angle than terrestrial laser scanning (TLS).

Terrestrial laser scanning provides plot-level observations between forest cover and snow distribution that can be used to validate UAS, airborne (e.g. Currier et al., 2019), and spaceborne lidar and confidently upscale local-scale processes (Revuelto et al., 2016a,b). Complimentary to nadir observations, TLS provides data collection with viewing angles from the ground and thus can capture fine-scale vegetation structure (and corresponding snow depth variations). Many studies have used TLS data to validate snow depths and melt (e.g. Deems et al., 2013; Hartzell et al., 2015; Grünewald et al., 2010), and along with physical modelling, to understand the role of wind in snow accumulation (e.g. Schirmer et al., 2011). While fewer studies have used TLS to explore forest canopy – snow relationships, TLS provides exciting opportunities to investigate fine-scale processes controlling snow distribution (Revuelto et al., 2015, 2016a, b; Gleason et al., 2013). Revuelto et al. (2015, 2016b) found smaller snow depth differences between the canopy and open areas in regions of thicker snowpack using TLS. They also demonstrated that shallower snow (snow depth < 0.5 m) occurred closer to the trunks while deeper snow (snow depth > 0.5 m) was found at the edge of the canopy where the dominant species was *Pinus sylvestris*. Gleason et al. (2013) used TLS to map tree stem density in burned forests and related this to greater snow accumulation in comparison to unburned areas. Taken together, previous studies point to the importance of choosing proper scales to study the controlling processes on snow depth variability; and furthermore, the opportunities to explore relationships between snow depth and canopy structure at fine-scales (individual trees).

The objective of this study is to further contribute to the understanding of fine-scale forest canopy – snow interactions by exploring how forest canopy structure affects snow depth distribution during the snow accumulation period with TLS. This study is part of the NASA-led SnowEx 2017 campaign aimed at evaluating remote sensing snow properties with a primary focus on testing the impact of forest on remote sensing approaches for monitoring SWE. We use TLS data collected during the accumulation period (single measurements) in mid-winter SnowEx 2017 (winter 2016-2017) across a number of small TLS study sites on Grand Mesa, CO. In this study, we explore the following questions:

1.      What measures of vegetation best describe a relationship with snow depth under the canopy (sub-canopy)?

2.      Are there conditions in which vegetation characteristics are a more important control on snow depth than topography, or vice versa?

3.      Does snow depth vary as a function of distance from the canopy edge? How does tree height influence snow depth as a function of distance and direction?

## 2 Study Area

The TLS data were collected at six sites (A, F, K, M, N, and O) across Grand Mesa, Colorado, USA. Grand Mesa is an approximate 470 km$^2$ plateau with elevation of 2,922 to 3,440 m, rising along a west to east gradient (Fig. 1). Vegetation in the west, where wind speeds are highest, is predominantly shrubby cinquefoil (*Dasiphora fruticosa*) steppe with isolated Engelmann spruce (*Picea engelmannii*) tree islands. The central portion of the mesa becomes semi-continuous forest cover consisting primarily of Engelmann spruce with minor subalpine fir (*Abies lasiocarpa*) and aspen (*Populus tremuloides*) trees,

all interspersed with subalpine meadows. Farther to the east, where wind speeds are lowest and elevation drops, there is dense continuous Engelmann spruce and subalpine fir forest with some lodgepole pine (*Pinus contorta var. latifolia*) and aspen stands at the lowest elevations.

Wind speed data during our TLS data collection period were available from three stations including a site at the western extent of forest cover on the plateau (Mesa West, near site A), a site termed the Local Scale Observation Site (LSOS), and a site

situated in more dense forest in the middle of Grand Mesa (Mesa Middle, near site M) (Houser, personal comm.) (Fig. 2). The data were collected from 17 November 2016 – 28 February 2017 and indicate a dominant NE wind direction at site A, though up to 15 m/s wind speeds from the SW were observed at this site. The predominant wind direction was from the NW at LSOS, and from the NW and SE at site M, during the sampling period. In analyses outlined below, we utilized a general E-W direction for testing the importance of wind (whereas we used a N-S direction for testing shading effects on snow, more below).

## 3 Methods

### 3.1 Data and Processing

We collected TLS data in snow-off (fall 2016) and snow-on (winter 2017) conditions at Grand Mesa at several sites (Fig. 1, Table 1) (Glenn et al., 2019; Hiemstra and Gelvin, 2019). The winter 2017 data collection occurred over 16 days but without significant snowfall between days. Each site was scanned once during the duration. A Riegl VZ-1000 (1550 nm) and Leica

Scan Station C10 (532 nm) were used. Multiple scans (at least 3) were obtained at each site to reduce occlusion. The scans were coregistered to produce a single point cloud for each site and date. Coregistered scans were then georegistered using surveyed locations within the plots. The georegistered scans (i.e. area of analysis) for each site ranges from approximately 10,000 to 38,000 m$^2$ (Table 1).

The TLS data were then utilized to derive snow depths, vegetation metrics, and topographic indices. From these data, we

utilized multiple linear regression to investigate relationships between the canopy and snow depths under the canopy at each of our sites. Snow depth relationships with topography in open areas with no trees (of a least 0.5 m height) were investigated using decision tree regression. Methods on identifying individual trees, under the canopy and in the open, are described below. The TLS point clouds were classified into ground or vegetation (fall 2016 dataset) and snow or vegetation (winter 2017 dataset), and then used to estimated snow depths at each of the sites, using several sub-routines in CloudCompare (v2.11 alpha;

retrieved from http://www.cloudcompare.org/). The TLS data were also used to for individual tree segmentation and to extract vegetation parameters using R 3.5.3 (R Core Team, 2019), lidR (v3.1.1; Roussel) and rLiDAR (v0.1.1; Silva) packages. These steps are outlined in Fig. 3.

While we did not independently assess the snow depth accuracy of the TLS data, Currier et al. (2019) assessed the relative accuracy of the same TLS data to airborne lidar at two of our sites. They indicate that the median snow depth difference
between the datasets (TLS and airborne) at sites A and K was less than 5 cm.

### 3.1.1 Ground, snow, and vegetation classification

The CANUPO method in CloudCompare was utilized to separate vegetation from ground and snow returns. This method includes training and classification. In the training step, we used 10,000 snow and vegetation samples to construct the classifier. We trained the algorithm at 15 different scales to assign features related to each class and selected the 9 best combinations of
scales (0.1m, 0.2m, 0.25m, 0.5m, 0.75m, 1m, 2m, 3m, 5m) to properly separate different classes. The combination of information from these scales helped the algorithm detect the dimension of each feature and assign snow and vegetation labels to the unclassified point clouds (Lague et al., 2013). CANUPO misclassified snow data points near tree stems as vegetation, and thus we reclassified these points manually using the software TerraScan (Helsinki, Finland). Manual classification included visually separating snow under the trees from tree trunks.

### 3.1.2 Snow depth estimation

The M3C2 algorithm (Lague et al., 2013) in CloudCompare was used to estimate under-canopy and open-area snow depths. In this algorithm, for every single point in the ground point cloud, a cylinder was defined with a range of different radii (projection scales) varying from 10 cm to 3 m and a length (height) of 3 m (see Lague et al., 2013, for details on these parameters) (Fig. B1). The orientation of the cylinder was along the normal vector of planes fitted on the ground points within
a 10cm radius. We projected all points within the cylinder onto the cylinder axis, took the vertical distance between projected snow, and ground points as the snow depth estimation. Through iteration, we found a balance between including enough TLS points for subsequent analysis and the accuracy of the snow depths (assessed with standard deviation) by using a 1 m projection scale. The resulting snow depth measurement has a relative accuracy of approximately 2.5 cm based on the maximum standard deviation from M3C2. Utilizing these measurements, we compared snow depths under the canopy and in the open at each site.
We also defined a transition zone as a 10 m buffer beyond each tree polygon in the direction of the open to identify any relevant differences within this zone.

### 3.1.3 Individual tree segmentation

We developed a canopy height model at 0.5 m resolution and identified tree tops to segment individual trees in the R package lidR. A local maxima was detected to identify tree tops using window sizes ranging from 1-3 m and minimum tree heights
from 2-6 m, depending upon the site. For areas with lower tree heights (0.5 – 2 m), we tiled the data that contained these trees

and segmented them in a similar approach. This allowed us to more accurately segment distinctly shorter and taller tree populations within sites, by adjusting segmentation parameters that worked better for those areas. Based on our preliminary analyses, the Marker-controlled watershed segmentation algorithm was most accurate (compared to li2012, dalponte2016, and watershed, all available in the lidR package). In addition, in cases where tall and short trees are close to each other, the algorithm could not detect shorter trees with large crown radii. Conversely, the branches far from the top of the tree may be considered as an individual tree if a small crown radii is used. This problem was resolved by tiling the las files, processing each separately, and then combining the results. An example of the segmentation results from site F is shown in Fig. B2. A similar process was performed for all sites.

To define *under the canopy* and *in the open*, we first performed segmentation to identify individual trees. Under the canopy was defined by all snow depth points within the tree polygons. To define the open area, we merged individual tree polygons that were less than 3 m from each other (patches of trees) and used the remaining areas as open. Site A was the only site dominated by shrubs (Fig. 1, Table 1) and we considered the shrub area as open (we removed shrubs in the processing and retained the ground points below) at this site because the focus of our study was on tree-snow relationships.

### 3.1.4 Vegetation and topography

We computed three vegetation metrics (Fig. 4) for each individual tree identified in the segmentation process (Table A1). These were then utilized, along with topographic metrics, to predict snow depths at each site using a multiple linear regression. Trees with at least 50 % snow cover below the tree crown (based on the segmented tree polygons) were used for analysis. The metrics included foliage height diversity (FHD), crown volume and the cumulative percentage of vegetation returns (zpcum). FHD represents the complexity of multi-layered vegetation structures (Clawges et al., 2008, Simonson et al., 2014). Trees with lower FHD have a lower number of layers, and thus less interception with snow. Crown volume describes the overstory cover of individual trees and is estimated by multiplication of crown surface area and crown height. Studies have shown overstory cover is negatively related to snow depth under the canopy (Hanley et al., 1987). Note that high crown volume does not necessarily equate to high FHD. The cumulative percentage of vegetation returns assessed across multiple layers within a tree allows us to understand whether a tree (as a whole) or a specific layer (cumulative) of the tree crown controls snow accumulation. We used 10 layers, starting at the bottom of each tree crown. In our preliminary analyses we used the first cumulative layer that explains the majority of vegetation returns within the crown (the first layer in which zpcumx > 50 %). The zpcum4 met this threshold at Site A and zpcum5 met this threshold for all other sites. By using these metrics, we are able to examine the effects of structural complexity (FHD), a specific cumulative layer (zpcumx) within the canopy, and/or the crown volume as a whole on snow depth.

Topographical metrics like elevation, eastness (sin (aspect) × slope) and northness (cos (aspect) × slope) are possible controls on snow depth for both under the canopy and open areas. We assumed that eastness represents the effect of wind based on the

predominant wind direction at the study site, and northness expresses the effect of solar radiation on snow. Slope and aspect were derived for each site using a nearest neighbor method in Arc Map 10.4.1 (ESRI, 2015) at 1m grid resolution.

To investigate the collinearity amongst and between the vegetation and topographic metrics, a variance inflation factor was computed. The variance inflation factor was close to 1 for all metrics at all sites except site O. The metrics were standardized to make sure that the scale of the independent variables did not affect the regression. We then used the vegetation and topographic metrics for each site (except site O) in a multiple linear regression model to assess their effect on snow depth under the canopy.

At site O and in the open areas, we utilized a decision tree regression in lieu of the multiple linear regression model. In the open areas, we examined the effect of elevation, northness and eastness on snow depths. Splits in the decision tree continued until the model could not improve beyond a $R^2 = 0.001$. To avoid overfitting, we validated the model by a range of complexity parameters (from 0.001 to 0.2) and pruned the tree by choosing the one with the smallest cross validation error. We trained the tree using 70 % of the data and validated the model prediction using 30 % of the data.

**3.1.5 Influence of canopy edge on snow depth**

Individual trees were used to assess snow depth variation at distances of 1 to 10 m away from the canopy edge. This represents how snow depth changes from the edge of individual trees to the open within a 10 m distance from the edge. We subsampled our data to only include trees that had good snow coverage (from TLS) within the buffer. This was determined based on the area around the tree having at least 50 % snow cover (see above).

Snow depth variation from the edge of trees had both increasing and decreasing trends. Thus, we split the data between increasing and decreasing snow depth trends and fit a model to each. We standardized the snow depth for each 1 to 10 m interval of individual trees. This allowed us to investigate all of the changes in snow depth across the same scale. We fit logarithmic (*standardized snow depth = Intercept + Coefficient ×log (distance)*) and linear (*standardized snow depth = Intercept + Coefficient × distance*) models to the increasing and decreasing snow depths, respectively, for both individual sites 215 and all sites together.

**3.1.6 Gap distribution and directional analysis**

We explored whether any of our sites were suitable for understanding the role of forest gaps (i.e. shading, interception) on snow depth distributions. While our study was not designed to analyze a range of gap distributions, the inherent forest density 220 and distribution gradient that spanned our sites across Grand Mesa provided this opportunity. In particular, we sought to identify if sites had a dispersed tree pattern, such that the gaps were large enough to prevent canopy interception of snow, and thus accumulated deeper snow. Seyednasrollah and Kumar, (2014) used a relationship of tree height and gap radius for evaluation of net radiation. We derived a similar but simplified gap distribution approach (Equation 1). We calculated the average distance of 10 nearest trees to each individual tree. This gave us a rough estimate of a gap size around each tree (D).

In the next step, we divided that average distance (D) by the average height (H) of those 10 nearest trees (D/H). This ultimately provides a ratio by which we can investigate the impact of shading from trees on gaps combined with gap size.

$$\frac{D_j}{H_j} = \frac{\frac{1}{k}\sum_{i=1}^{k} d_{ij}}{\frac{1}{k}\sum_{i=1}^{k} h_{ij}} \tag{1}$$

Equation (1), illustrates the gap distribution for an individual tree (j) where, $D_j$ is the mean distance of the k closest trees to tree j; $H_j$ is the average height of k closest trees to tree j; k is the number of neighbors and $d_{ij}$ and $h_{ij}$ are the distance and height of tree i to tree j, respectively.

Secondly, we performed an average nearest neighbor analysis of the distribution of trees at each of the sites. In this analysis, we tested for tightly clustered trees in which gaps were minimal (clustered), randomly distributed trees where gaps could potentially lead to deeper snow accumulation (random), or dispersed trees where no particular pattern exists and thus gaps are likely not prevalent (dispersed).

We also investigated whether relationships between tree heights and snow depth variation are significant based on direction. We did this using the 10 m transition zone (buffer) for each individual tree. We classified snow depths within each buffer in the four cardinal directions and fitted a linear ($snow\ depth = \alpha \times (tree\ height) + \beta$) or nonlinear ($snow\ depth = \alpha \times exp(\beta \times tree\ height) + \theta$) model, depending upon the site, between tree heights and mean snow depth per each direction. We also performed a directional analysis with a Wilcoxon signed-rank test for comparing snow depth on the north and south sides (and east and west) for individual trees at each site. Note that due to sampling extents, our transition zone analysis was performed at 1 m increments instead of at multiples of mean tree height as in previous literature (e.g. Currier and Lundquist, 2018).

## 4 Results

### 4.1 Snow depths

Using our individual tree analysis, we found higher snow depths (1-1.6 m on average) in open regions and lower snow depths in areas dominated with trees (0.8-1.3 m on average) (see Figs. 5 and 6). Snow depths were 12-28 % higher in the open than under canopy. Mean snow depth percent change between the 10m transition zone and under the canopy ranges from less than 1 % for sites A and K to a maximum of 7 % at site M. We found the lowest mean snow depths in our most westerly site (A), which is dominated by dense clusters of relatively rigid shrubs (*Dasiphora fruticosa*) and has the lowest tree cover of all sites.

The standard deviation (SD) of snow depths was similar between the transition zone and under the canopy for four sites (A, F, K, and O). We found a lower SD of open area snow depths at site O compared with under-canopy and transition zones (Fig. 6).

## 4.2 Influence of vegetation and topography on snow depth under the canopy

A multiple linear regression model was applied to assess the effect of vegetation and topographical metrics on snow depth under the canopy. The  regression explained 43, 54, 27, 25 and 28 percent of snow depth variation at sites A, F, K, M, and N respectively (Table 2). Based on the models, FHD was the most influential vegetation metric at five sites (Table 2). Figure B3 shows the distribution of FHD at each of the sites, with higher FHD demonstrating more evenly spaced foliar arrangement along an individual tree. Most of the sites had two peaks of FHD distributions. FHD and snow depth were negatively related at all sites, i.e. a vertically-sparse foliar arrangement resulted in higher snow depths. At site A, a negative relationship (-0.21) between cumulative percentage of returns within the fourth layer (zpcum4) and snow depth occurred. FHD also showed a higher negative relationship at this site (-0.27) with snow depth. The results indicate that the effect of eastness, northness and crown volume was not significant (p-value > 0.001) and elevation positively affected (with a coefficient of 0.14) the snow depth under the canopy at this site. At site F, elevation and FHD were the most important features that explained 54 % of snow depth variation. Vegetation and topography could not explain more than 30 % of snow depth variance under the canopy at sites K, M and N.

Because of collinearity between eastness and northness at site O, we used a regression tree to investigate the effect of different features on snow depth under the trees. Eastness and FHD were the most important features at site O respectively (Fig. 7). A decision tree regression for this site explains 74 % of snow depth variance under the trees. The results also indicate as we move from east (positive) to west (negative), the snow increases with higher slopes. In other words, larger, west-facing slopes are covered by deeper snow. In addition, shallower snow depths are predicted in the canopy with higher FHD at this site.

## 4.3 Influence of slope, aspect and elevation on snow depth in open areas

We examined the effect of topography on snow depth in open areas using a decision tree regression for each site. Based on the regression tree (Table 3), elevation was the most important feature at sites A, F, K and was the second most important feature at site M and N. Decision trees could predict 38, 36, 36, 31, 18 and 64 percent of snow depth variations at sites A, F, K, M, N and O respectively. The model slightly overfitted for sites A, M and N. However, the $R^2$ for the training and testing datasets at the other three sites are similar. Eastness and northness represent wind and solar radiation impacts on snow depth variation with regard to topography. Except site O where topography (eastness and northnes) explained 64 % of snow depth variation, topographical metrics could not explain more than 38 % of snow depth at the other sites (Table 3). This is likely a result of scale, in which our plot sampling size did not adequately sample the effects of topography and wind on snow depth variation. We found that at site A, elevation and northness were influential on snow depths in the open. High snow depths were found in open northeast facing slopes (same as predominant wind direction) at site A (see Figs. 1, 8b). Site O was the only site that we found an influence of both eastness and northness on snow depth. The influence of eastness occurred in both the open and under the canopy (Table 3, Figs. 7, 8a, b). This site has high north and west facing slopes (in both under canopy and the open) with relatively higher snow depths; whereas south facing slopes have relatively lower snow depths.

### 4.4 Influence of canopy edge on snow depth

We found that snow depths increase with distance from the canopy edge into the open for the majority of individual trees (Figs. B4-9). However, at some sites we found a decreasing snow depth trend by moving farther from tree edges. For example, this occurred on the northwest side of the tree patches in the southeast portion of site O. This is the area of site O where the northwest facing slope has likely the largest influence on snow depths. The increasing snow depth trend from the canopy edge occurs in the north where snow depths are low (less than 1 m). Site A also showed a decreasing snow depth pattern in the north/northwestern sampled region, and this is likely due to northeast winds and deeper snow depths in the northeast facing slopes in the southern portion of the site.

Results show that a logarithmic regression can explain more than 85 % of an increasing snow depth trend at each site. Figure 9a illustrates the logarithmic regression between snow depth and distance from the edge for all sites together. The model coefficients were almost the same for individual sites as well as all sites together (Table 4). This indicates that within a 10 m distance from the edge of the trees, snow depth increases with a unique logarithmic trend.

For decreasing snow depth, a linear regression explained 72 % of snow depth variation at all sites together. However, snow depth decreases at site K followed a second order polynomial, which covers only 4 % more variation than simple linear regression. The coefficient and p-values for linear regressions are illustrated in Table 5. Figure 9b also shows the regression fit between decreasing snow depth and distance from the tree edge.

### 4.5 Gap distribution and directional analysis

Our results show that site N has the largest median D/H ratio (0.74) compared to all other sites of <0.5 (Table 1). Site N is the only site with a randomly dispersed tree pattern (Table A2) and thus the most likely site to experience lower interception, possibly resulting in deeper snow.

We found a negative relationship between tree heights and snow depths based on direction at sites A, K and O (Table A3, Fig. 10). Snow depth decreased exponentially at site A with an increase in tree height. However, this relationship was linear for sites K and O and was not significant for the other three sites. An exponential fit could explain 56, 61, 76, and 32 percent of snow depth change on north, west, south, and east directions at site A, respectively (Table A3, Fig. 10a). We found snow depths were different between the north and south sides of trees at sites A, K, and O but not for any other sites or directions (Table A4).

## 5 Discussion

We observed several interesting relationships between vegetation canopies, topography, wind and snow depths across our sites. As expected, snow depths were deeper in the open compared to under canopy. However, describing the relationships between vegetation and snow is complicated by the structure, distribution (pattern), and type of vegetation. The relationship is further convoluted by local topography and wind speed/direction. For example, we found that slope, aspect, and wind (rather

than vegetation) might control snow depths at local scales at two of the sites, A and O. This is not surprising, as site A was dominated by 0.4-0.6 m tall shrubs and wind exposed, and site O had a relatively low tree canopy cover. While site A had the lowest tree canopy cover in our dataset, we only sampled the edge of a much larger patch of trees (based on field observations). Our results indicate that local topographic interactions with wind have a major influence on snow accumulation, especially when we do not consider the much larger landscape controls. While sites A and O have slopes within the overall range of all of our sites, the combination of local slope and aspect for site O, appear to be driving factors in snow depths. In fact, site O has the highest mean snow depth (1.44 m), likely due to these local site conditions.

When our analyses were confined to under the canopy of individual trees, we generally found a significant relationship between the vertical spatial arrangement of the foliage (based on the FHD) and snow depth, but this relationship did not hold across all sites. For example, FHD and northness explained 25 % of the variance of snow depth at site M. This site has the highest mean tree heights of the study. Taken together with northness as the most important feature at this site, solar radiation likely had higher control on snow depths than on the particular foliar arrangement of the trees at this site. Overall, sites M and N had the least effect of vegetation metrics on snow depth. This may be due to the vegetation patterns at these sites (under the canopy slope and aspect have no effect at site N (Table 2, Fig. 8a)). Site M is a relatively open area with mature Engelmann spruce and subalpine fir trees in the SW and NE areas of our site. Subalpine fir trees are generally more slender than Engelmann spruce, and thus their shape may not be as influential on accumulation of under canopy snow depths. Site N has the highest percent cover and the smallest trees (mean tree height 10.5 m, SD of 2.62 m, Table 1). This second growth canopy is the only site dominated by lodgepole pine, which are also slender. While the mean FHD is similar to the other sites, site N is the only site with trees in a dispersed pattern in which the size of the gaps likely prevents snow interception, and thus provides an opportunity for snow accumulation. In fact, site N had the second highest mean snow depth under the canopy (1.38 m, compared with 1.44 m at site O, Figs. 6 and 8a). Testing for a dispersed tree pattern could be beneficial to future studies, especially because previous research (e.g. Ning Sun et al., 2018) found gap size to be a control on snowmelt timing; however, our study was during the accumulation phase so we cannot draw similar conclusions at site N.

Our models show that FHD is the only control on snow depth at site K, explaining 27 % of snow variation under the canopy. The standard deviation of elevation at this site is less than the other sites (Table 1), likely indicating that snow depth under the canopy is not affected by topography. This indicates that in the absence of large changes in topography, snow depth under the canopy is most likely controlled by trees. In the open, elevation and eastness are predominant controls on snow depth. The latter is perplexing because a wind effect was not observed under the canopy (Table 2) nor along east-west sides of trees (Table A4). In the absence of other information, we conclude that the spatial arrangements of the trees may shelter the open area, causing higher snow depths with no interception. Elevation and FHD at site F explain 54 % of snow depth variation, which is the greatest among the sites. The site's large standard deviation of elevation (Table 1) explains why elevation is the most important control in both under the canopy and in open areas. The predominant wind direction is from west to east at the nearby LSOS station. However, our models indicate that wind has little influence on snow distribution, and rather solar radiation along with elevation control snow patterns in the open at our scale of analysis.

Our canopy edge analyses generally found that as the distance increases from the canopy edge, snow depths also increase. This finding is congruent with previous studies which used airborne lidar across larger spatial extents and lower vertical resolutions in the canopy (e.g. Moeser et al., 2015a; Mazzotti et al., 2019). We leveraged the fine-scale observations in our dataset to model snow depth increases from the canopy edge, and found the trend to generally follow a logarithmic model at individual and all sites together. In contrast to our observations, Hardy and Albert (1995) indicate that snow depth changes uniformly

from the tree edges towards the open. However, the patterns we observed in our TLS data are similar to those Mazzotti et al. (2019) observed from airborne lidar at Grand Mesa, over the same time frame. While we observed snow depth to decrease nonlinearly with distance from canopy edge at one site, a linear decrease was the norm amongst sites and was likely mediated by wind and/or topography feedbacks at sites A and O. In summary, we expect snow depth increases from the canopy edge toward the open, but wind and topographic controls may affect this trend.

We did not find high snow depth accumulation or variability within a transition zone similar to the findings of Broxton et al. (2015). While their study included similar tree species, wind speeds and elevation, their spatial scale of analysis was larger with the use of airborne lidar. The relation between tree height and snow depth in cardinal directions from individual trees indicate that we expect shallower snow within the 10 m transition zone from taller trees (Fig. 10). In other words, two adjacent trees with different heights affect snow depth differently in any one direction and shorter trees generally deeper snow in all

directions. We expected the opposite to occur i.e. taller trees should create larger shadows and provide more shading/sheltering. As our snow-on datasets are from the accumulation season, we may not see shading effects of taller trees in the transition zone; negligible melt had occurred at the time of these surveys. If our datasets extended throughout the season (our data is a single measurement in time), we might expect these relationships to change.

    Following previous studies that show a directional relationship with snow depths (e.g. Mazzotti et al., 2019; Currier and

Lundquist, 2018), we found significantly different snow depths between the north and south sides of trees at site A, K, and O. This may be due to the local topography and wind at sites A and O. Additionally, previous lidar-based canopy snow interaction studies (Trujillo et al., 2007, 2009; Deems et al., 2006) relied on simple canopy models using maximum height. Our results show that in nearly all situations, structural information contained in denser lidar point clouds have more predictive capability.

    A limitation of our study is that the results are site specific and cannot be generalized to all forest conditions. In addition, our

data are best suited to fine scale interactions between individual trees, or clusters of trees, and under the canopy or surrounding snow depths. Data gaps may exist from occlusion within dense canopies. We minimized the effect of occlusion by performing our analysis on individual trees. Notably, the directional analysis is not affected by occlusion as we used tree height in the analysis and trees having at least 50 % snow cover. Occlusion with TLS can be eliminated using UAS and/or airborne lidar as their nadir/off-nadir scan positions can cover the canopies at different angles. However, mapping dense canopies and mapping

snow depth under dense canopies may still be a challenge. Ultimately, TLS provides data for investigating fine-scale controls, and is highly complementary to UAS and airborne lidar, which can help test larger scale features, such as gap area across space.

## 6 Conclusions

Our study indicates that even with fine scale, individual tree observations from TLS, vegetation structural metrics are not enough to describe snow depth during the accumulation season. Local scale topography and wind should also be considered. While our sites were not designed solely for intercomparison, we found notable trends in our site comparisons. The vertical arrangement of foliage (e.g. FHD) of individual trees influences under canopy snow depths, and in some cases, quite strongly. Whereas cumulative percentage of returns and crown volume were less powerful explanatory variables. Further studies should be designed to test this within and between species. For example, our sites were primarily Engleman spruce, subalpine fir, and lodgepole pine, all of which have different canopy structural shapes. Further studies targeting samples of each of these with different foliar arrangements and heights should be undertaken to fully understand the implications of FHD and tree heights on snow depths at local scales.

We also found that topography had greater control than vegetation at sites where slopes favored wind conditions for increasing snow depths, or where vegetation presence was minimal. While the latter may be obvious, increased observations with varying vegetation cover, wind, and topography should be considered with TLS.

This study highlights the complementary nature of TLS observations to UAS and airborne lidar, where TLS can provide fine scale observations within the canopy and relationships with under the canopy snow depth. Data from TLS can also be used to validate airborne lidar (e.g. Currier et al., 2019), and further studies should investigate how vegetation metrics such as FHD compare between TLS, UAS, and airborne lidar in these snow-dominated forest ecosystems. Further, along with UAS and airborne lidar, TLS provides a complementary dataset for upscaling to similar types of vegetation structure and topography observed from satellites such as the NASA Ice, Cloud and land Elevation Satellite (ICESat-2), or missions such as Global Ecosystem Dynamics Investigation (GEDI).

## Data availability

The TLS data are from 2017 SnowEx campaign collected at Grand Mesa, Colorado and are accessible at https://nsidc.org/data/SNEX17_TLS_PC_BSU/versions/1 and https://nsidc.org/data/SNEX17_TLS_PC_CRREL/versions/1.

## Author contribution

Zach Uhlmann, Lucas Spaete, Christopher A. Hiemstra, Christopher J. Tennant, Art Gelvin, and Nancy Glenn contributed field data collection. Lucas Spaete, Nancy Glenn, Christopher A. Hiemstra, Hans-Peter Marshall, and Art Gelvin designed the project. In addition, Zach Uhlmann, Nancy Glenn, Ahmad Hojatimalekshah, Christopher A. Hiemstra, Christopher J. Tennant, Jake Graham, Hans-Peter Marshall, Jim McNamara and Josh Enterkine contributed in writing and interpretations. Ahmad Hojatimalekshah did the analysis.

The authors declare that they have no conflict of interest.

## Acknowledgments

Funding and support for the project was provided by NASA Awards 80NSSC18K0955 and NNX17AL61G, NASA SnowEx 2017 campaign, and the Department of Geosciences, Boise State University. We would like to thank all those who participated and supported the NASA 2017 SnowEx campaign.

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

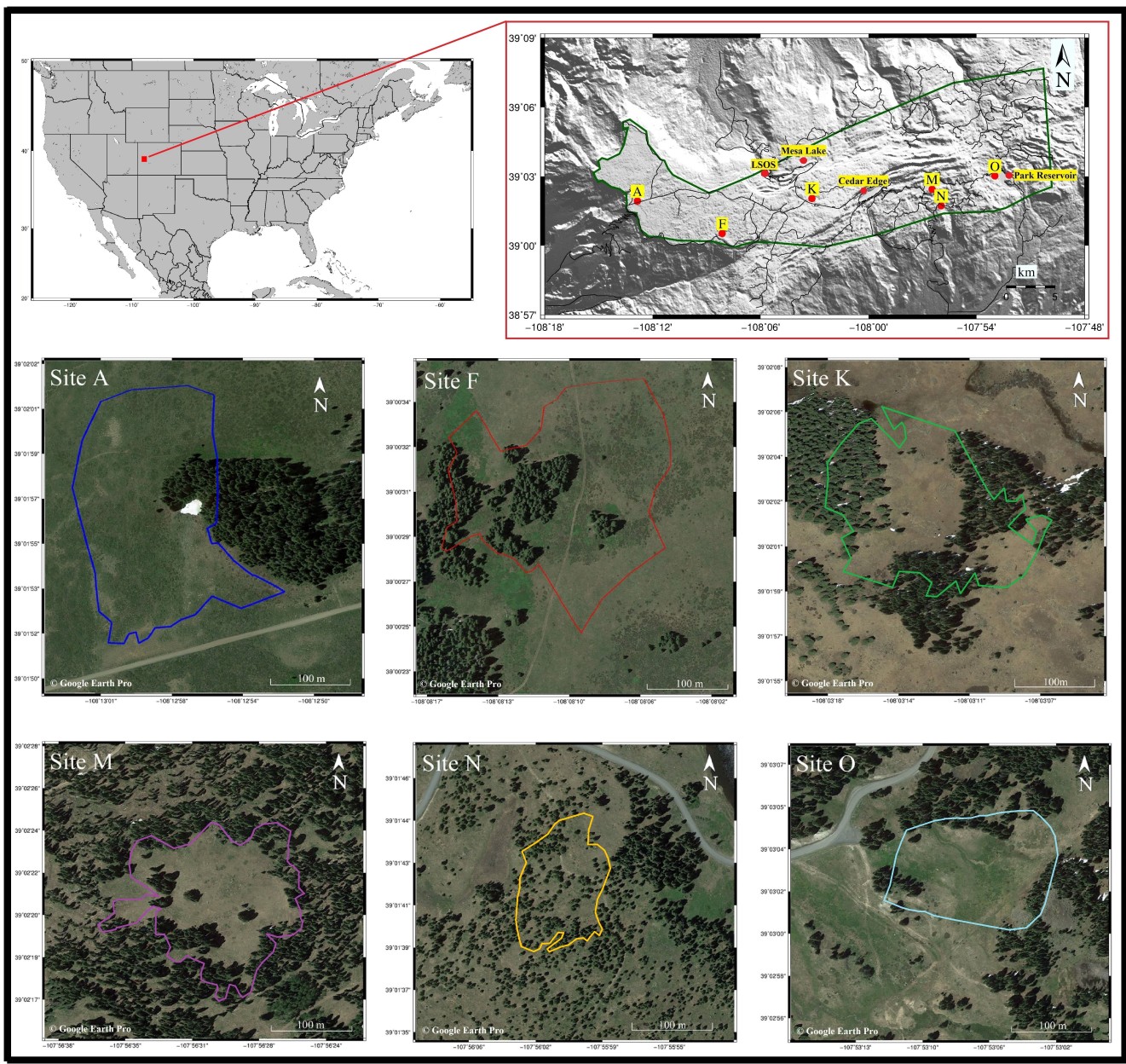

**Figure 1: Study area and location of TLS sites and meteorological stations.**

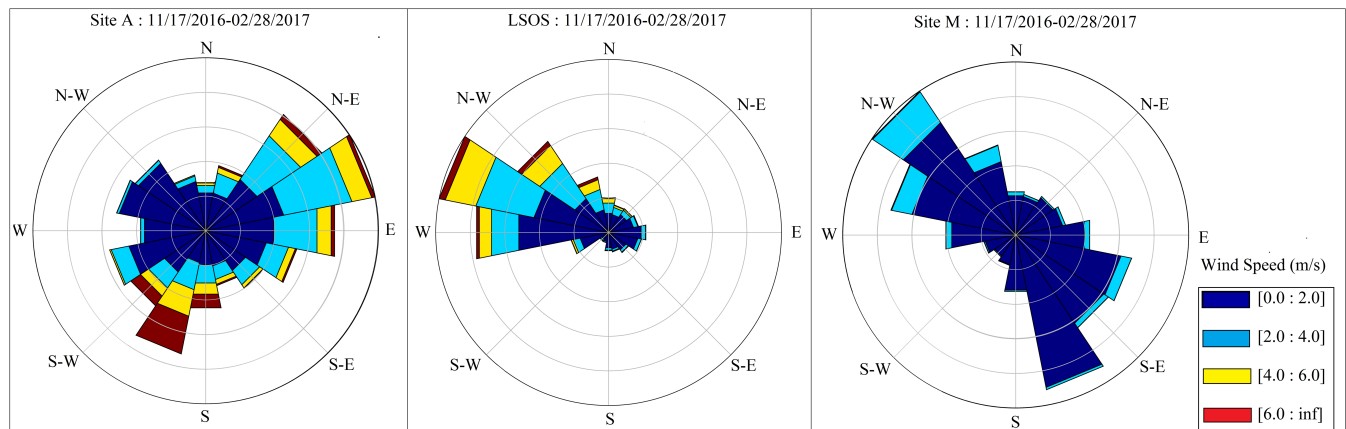

**Figure 2: Rose diagrams of meteorological stations A, LSOS, and M.**

**Figure 3: TLS data processing workflow.**


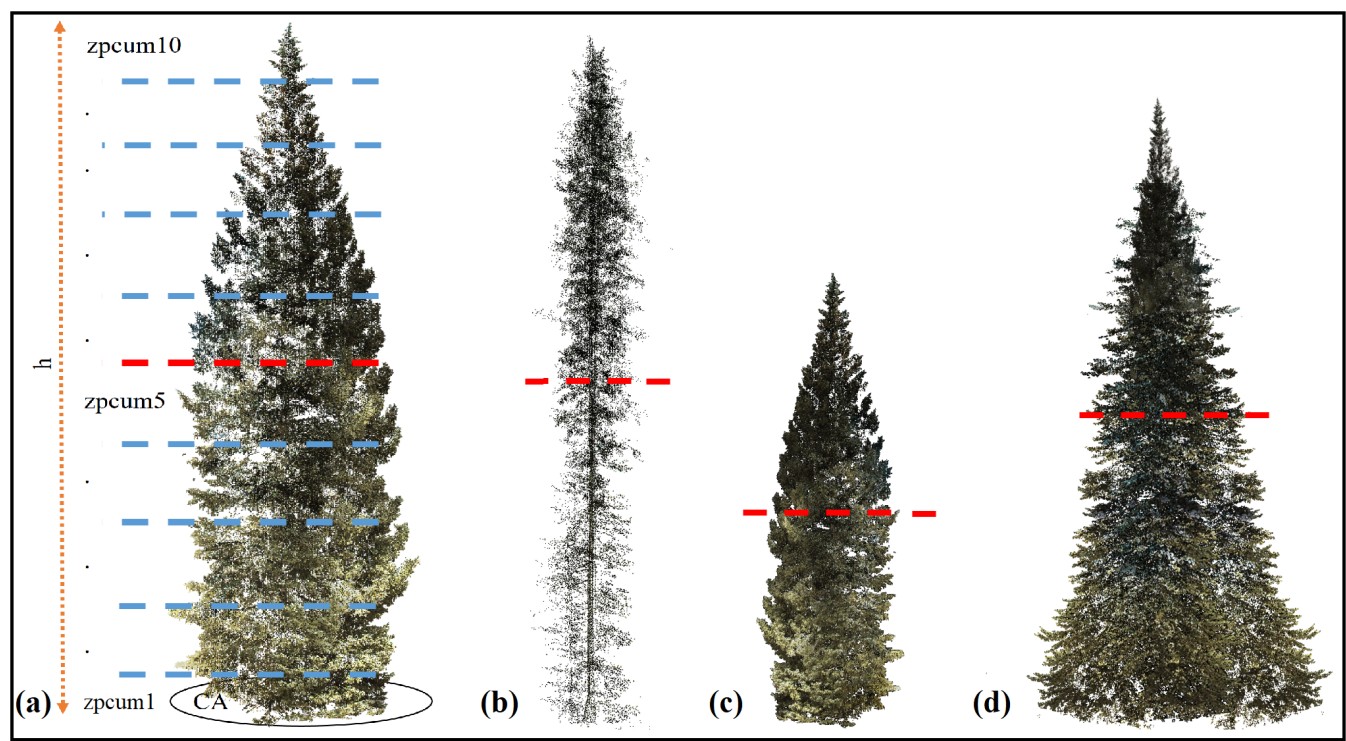

**Figure 4: Graphical examples of trees with different structures and the metrics of foliage height diversity (FHD), crown volume and the cumulative percentage of vegetation returns (zpcum). Crown volume is estimated using crown area (CA) x height (h). Zpcum is based on 10 layers (zpcum1–zpcum10) (see example in (a)). Red dashed lines are examples of the cumulative layer with > 50 % of vegetation returns. The foliar complexity (FHD) of (a) and (c) are similar but the crown volumes and cumulative percentage of vegetation returns are different; whereas the FHD of (b) is the lowest.**


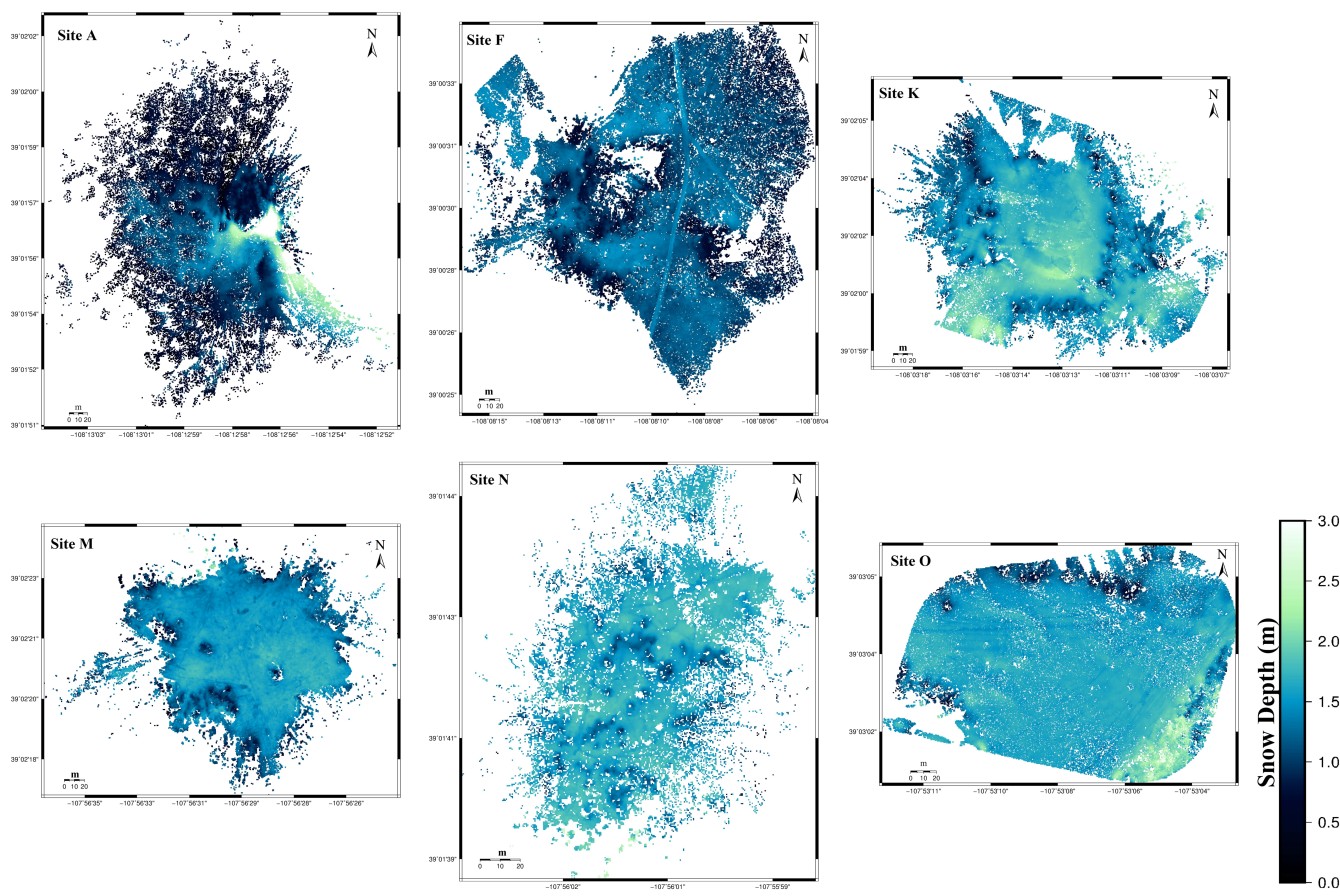

**Figure 5: Snow depths at each site from TLS data.**

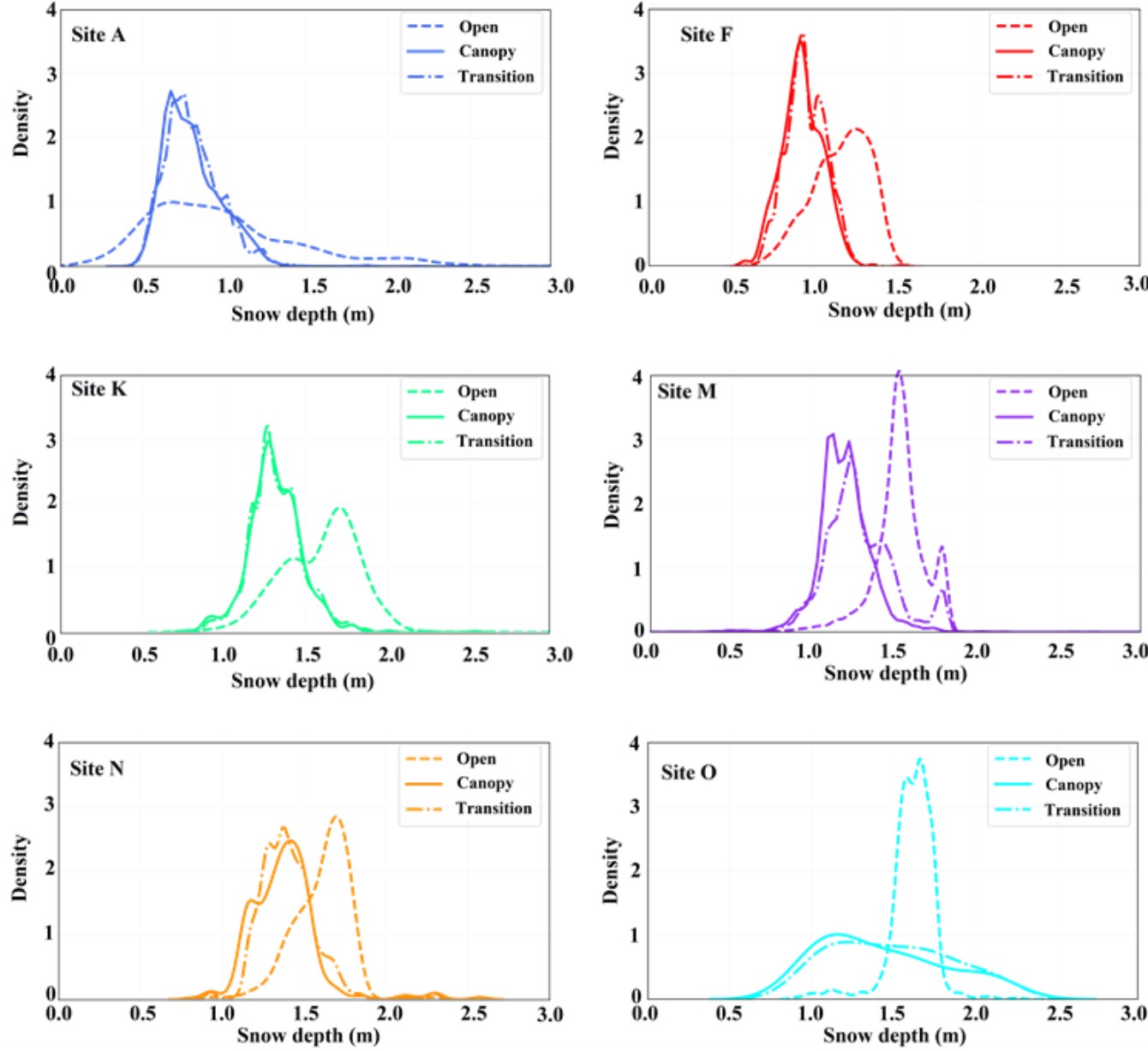

**Figure 6: Snow depth under the canopy, within the 10 m transition zone, and in the open.**

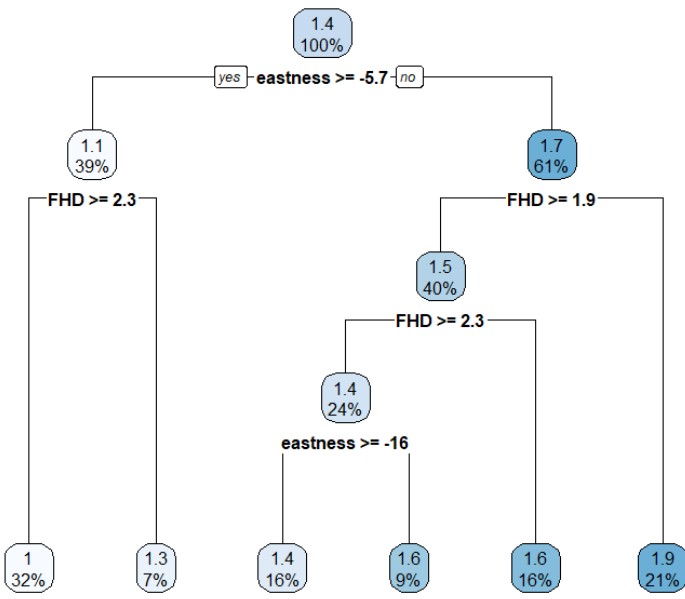

Figure 7: Decision tree at site O. Snow depth (m) are represented along with percent of the individual trees used for analysis.

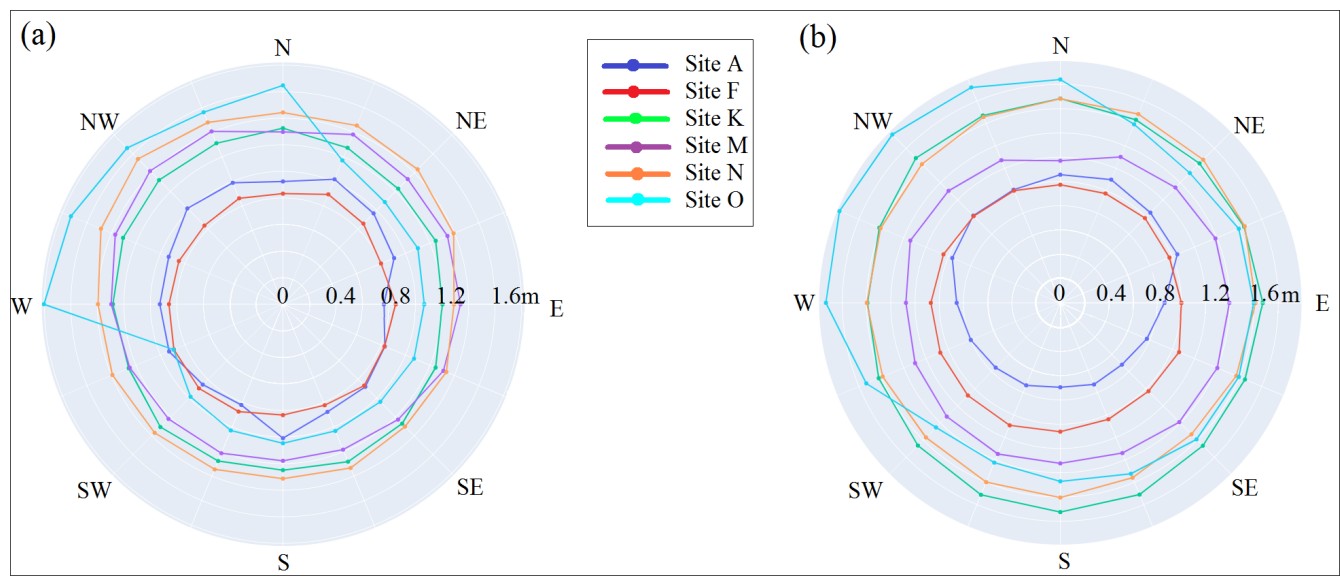

Figure 8: Under the canopy (a) and open area (b) snow depth at different aspects.

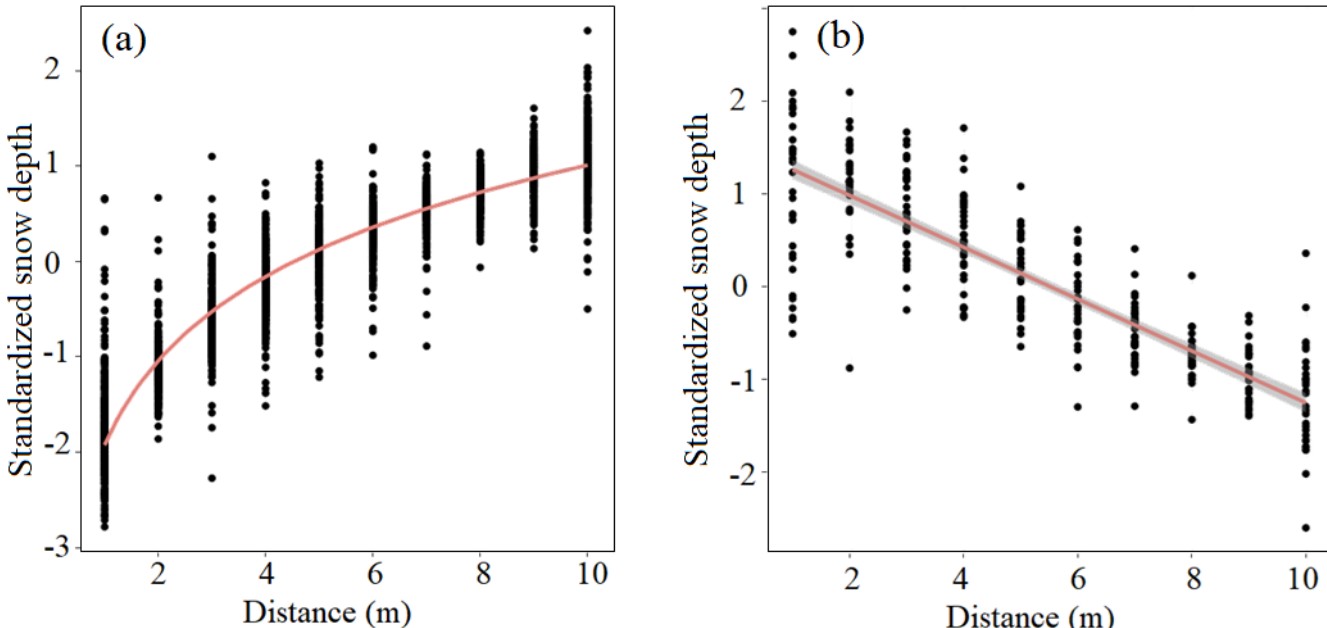

Figure 9: Increasing (a) and decreasing (b) snow depth trends and regression within a 10 m distance from the tree edges for all sites together. The confidence interval depicted in this figure is 95 %.

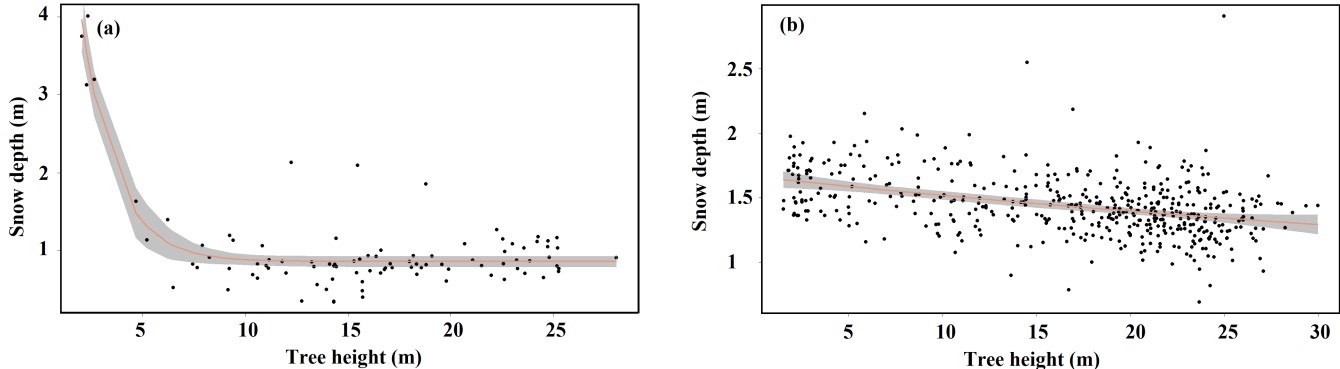

Figure 10: Tree height and snow depth within the 10 m transition zone in the south direction for (a) site A and (b) site K. The exponential and linear fit is shown with a red line along with 95 % confidence interval (grey).

**Table 1. Terrestrial laser scanning site descriptions. Sd = standard deviation. D/H = Distance/Height. FHD = foliar height diversity (described below).**

| Site | Area of Analysis (m²) | Mean Elevation (m) (sd) | Elevation Range (m) | Wind Direction | Vegetation Type | Tree % Cover | D/H | Range of Tree Height (m) | Mean Tree Height (m) (sd) | Median FHD (m) (sd) | Comments |
|------|------|------|------|------|------|------|------|------|------|------|------|
| A | 29128 | 3037 (3.6) | 3027-3045 | Westerly | Mostly Shrubs with Engelmann Spruce | 9 % | 0.36 | 2-27 | 16.6(5.77) | 2.27(0.48) | Sampled edge of larger tree cover extent. |
| F | 37838 | 3105 (6.6) | 3090-3118 | … | Engelmann Spruce | 24 % | 0.32 | 2-29.2 | 21.4(6.38) | 2.64(0.42) | Patchy Dense Trees. |
| K | 31497 | 3253 (1.4) | 3249-3257 | … | Engelmann Spruce | 38 % | 0.41 | 1.7-30 | 18.7(6.45) | 2.75(0.38) | Patchy Trees. |
| M | 21994 | 3122 (4.7) | 3114-3142 | NW to SE | Engelmann Spruce | 45 % | 0.39 | 5.7-33.7 | 21.6(7.45) | 2.7(0.46) | Patchy Trees. |
| N | 10187 | 3054 (2.4) | 3047-3061 | … | Lodge Pole Pine | 52 % | 0.74 | 1.7-28.3 | 10.5(2.62) | 2.77(0.42) | Second-growth, dispersed Lodge Pole Pine. |
| O | 24302 | 3067 (4.0) | 3055-3078 | … | Subalpine Meadow with Engelmann Spruce | 14 % | 0.39 | 5-33.4 | 16.4(7.65) | 2.82(0.37) | Two main slopes, one towards the south and other towards the NW. The later has the deepest snow. |

**Table 2. The first and second highest coefficients and their associated p-values for multiple linear regression between vegetation, topography and snow depth under individual trees. P-values > 0.001 are described as not significant.**

| Site | Highest coefficient | p-value | 2nd highest coefficient | p-value | Adjusted R-squared |
|------|---------------------|---------|-------------------------|---------|--------------------|
| A | FHD (-0.27) | <0.001 | Zpcum4 (-0.21) | <0.001 | 0.43 |
| F | Elevation (-0.08) | <0.001 | FHD (-0.07) | <0.001 | 0.54 |
| K | FHD (-0.11) | <0.001 | Not significant | Not significant | 0.27 |
| M | Northness (0.08) | <0.001 | FHD (-0.05) | <0.001 | 0.25 |
| N | FHD (-0.09) | <0.001 | Elevation (0.06) | <0.001 | 0.28 |

**Table 3. The first and second important features from decision tree regressed between topographical features and snow depth in the open areas.**

| Site | First important feature | Second important feature | Train $R^2$ | Test $R^2$ |
|------|-------------------------|--------------------------|-------------|------------|
| A | Elevation | Northness | 0.46 | 0.38 |
| F | Elevation | Northness | 0.36 | 0.36 |
| K | Elevation | Eastness | 0.39 | 0.36 |
| M | Northness | Elevation | 0.39 | 0.31 |
| N | Northness | Elevation | 0.30 | 0.18 |
| O | Eastness | Northness | 0.68 | 0.64 |

**Table 4. Logarithmic regression between distance from the tree edge and increasing snow depth within a 10 m buffer toward the open. P-values > 0.001 are described as not significant.**

| Site | Intercept | Coefficient | p-value | Adjusted $R^2$ |
|------|-----------|-------------|---------|----------------|
| A | -1.91 | 1.27 | <0.001 | 0.86 |
| F | -1.97 | 1.3 | <0.001 | 0.91 |
| K | -1.9 | 1.26 | <0.001 | 0.85 |
| M | -1.92 | 1.27 | <0.001 | 0.87 |
| N | -1.95 | 1.29 | <0.001 | 0.89 |

| | | | | |
|---|---|---|---|---|
| O | -1.9 | 1.26 | <0.001 | 0.85 |
| All sites together | -1.92 | 1.27 | <0.001 | 0.87 |

**Table 5. Linear regression between distance from the tree edge and decreasing snow depth within a 10 m buffer toward the open. Site F had no trees with a decreasing trend. P-values > 0.001 are described as not significant.**

| Site | Coefficient | p-value | Adjusted $R^2$ |
|---|---|---|---|
| A | -7.6 | <0.001 | 0.70 |
| F | … | … | … |
| K | -6.2 | <0.001 | 0.70 |
| M | -3.6 | <0.001 | 0.47 |
| N | -4.2 | <0.001 | 0.63 |
| O | -9.0 | <0.001 | 0.82 |
| All sites together | -14.6 | <0.001 | 0.72 |

**Appendix A**

**Table A1. Vegetation metrics derived for individual trees at each of the sites with equations and references.**

| Vegetation metrics | Equation |
|---|---|
| Cumulative percentage of return in the xth layer (zpcum*) | $$zpcumx = \sum_{i=1}^{x} \frac{\# \, veg \, returns \, (i)}{total \, veg \, returns} * 100$$ Where i is the ith layer of tree. (Reference: R 3.5.3 (R Core Team, 2019), lidR (v3.1.1; Roussel) package). |
| Foliage Height Diversity (FHD) | $$FHD = -\sum P_i \ln (P_i)$$ Where $P_i$ is the proportion of the number of lidar returns in the ith layer to the sum of lidar points of all the layers (using all points) (bcal lidar tools documentation) (Reference: BCAL Lidar Tools, Boise State University, Department of Geosciences, URL: https://github.com/bcal-lidar/tools). |
| Crown volume (crwnvlm) | (Reference: R 3.5.3 (R Core Team, 2019), rLiDAR (v0.1.1; Silva) package). |


**Table A2. Average nearest neighbour results for each site. The null hypothesis here is that trees are randomly distributed. The nearest neighbour ratio is the mean of observed mean distance over the expected mean distance between neighbours assuming a random distribution. The ratio equal or close to one is considered random. Only site N has a random distribution pattern of trees. As site A contains a dense canopy and several trees far from it, the method takes it as dispersed.**

| Site | p-value | Z-score | Nearest Neighbor Ratio | Tree pattern |
|---|---|---|---|---|
| A | 0.00 | 6.92 | 1.28 | Dispersed |
| F | 0.00 | -3.72 | 0.91 | Clustered |
| K | 0.00 | -7.60 | 0.85 | Clustered |
| M | 0.00 | -8.18 | 0.79 | Clustered |
| N | 0.28 | -1.09 | 0.97 | Random |
| O | 0.00 | -12.66 | 0.55 | Clustered |


**Table A3. Relationship between tree height and snow depth (within a 10 m transition zone) based on cardinal direction. The columns are site, adjusted r-squared, tree height range, and number of trees. Ranges of tree heights vary within sites because not all trees had adequate snow samples to test in all cardinal directions.**

| Site | North | Tree height range (m) | # of trees | West | Tree height range (m) | # of trees | South | Tree height range (m) | # of trees | East | Tree height range (m) | # of trees |
|---|---|---|---|---|---|---|---|---|---|---|---|---|
| A | 0.56 | 2.05– 25.52 | 95 | 0.61 | 2.05– 28.05 | 91 | 0.76 | 2.05– 28.05 | 94 | 0.32 | 2.05–28.05 | 101 |
| K | 0.16 | 1.51 – 29.34 | 486 | 0.11 | 1.51 – 29.34 | 485 | 0.15 | 1.51– 29.97 | 474 | 0.17 | 1.51–29.97 | 483 |
| O | 0.29 | 5.46– 33.42 | 122 | 0.51 | 5.09– 33.42 | 127 | 0.44 | 5.09– 33.42 | 133 | 0.4 | 5.46– 32.46 | 131 |


**Table A4: Wilcoxon signed-rank test results for comparing snow depth on the north and south sides (and east and west) for individual trees at each site. Note only trees that have snow depth data on both north and south or east and west are considered. Statistically significant are bolded.**

| Site | p-value (North-South) | Number of Samples | p-value (East-West) | Number of Samples |
|---|---|---|---|---|
| A | 0.000 | 59 | 0.350 | 60 |
| F | 0.807 | 138 | 0.704 | 130 |
| K | 0.035 | 390 | 0.780 | 397 |
| M | 0.650 | 170 | 0.780 | 163 |
| N | 0.720 | 202 | 0.495 | 195 |
| O | 0.024 | 90 | 0.430 | 94 |


**Appendix B**

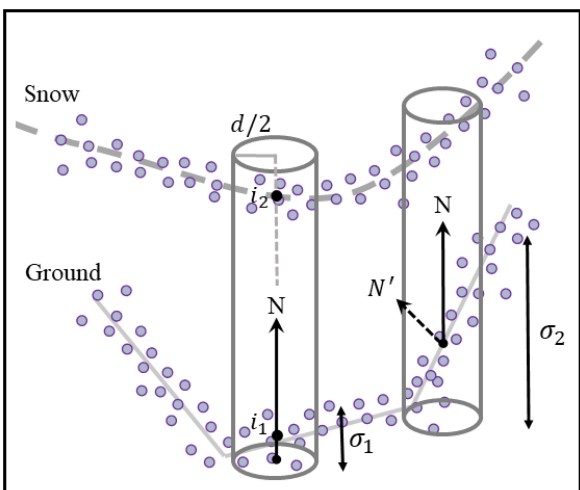

**Figure B1: M3C2 method for computing snow depth. Vector N shows the normal on the reference surface (ground) and d is the projection scale. Surface roughness is the standard deviation of the point clouds within the cylinder ($\sigma$). Misorientation on rough surfaces ($N'$) is seen by high standard deviation ($\sigma_2$) and is resolved by choosing the proper normal scale. Snow depth is the vertical distance between the average positions of ground and snow point clouds within the cylinder (distance between $i_1$ and $i_2$). Redrawn from Lague et al., 2013.**

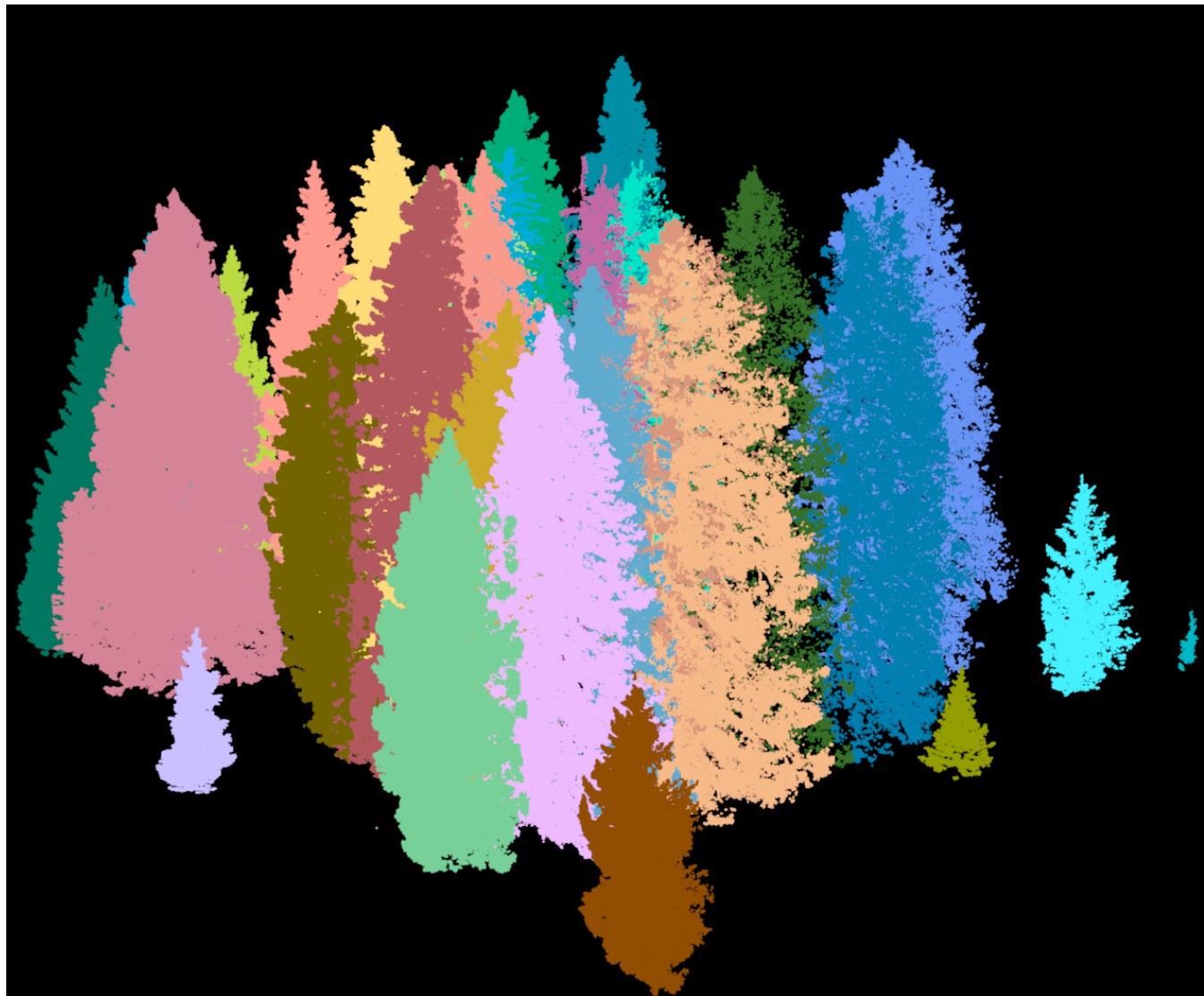

**Figure B2: As an example, segmentation results for one las tile at site F.**

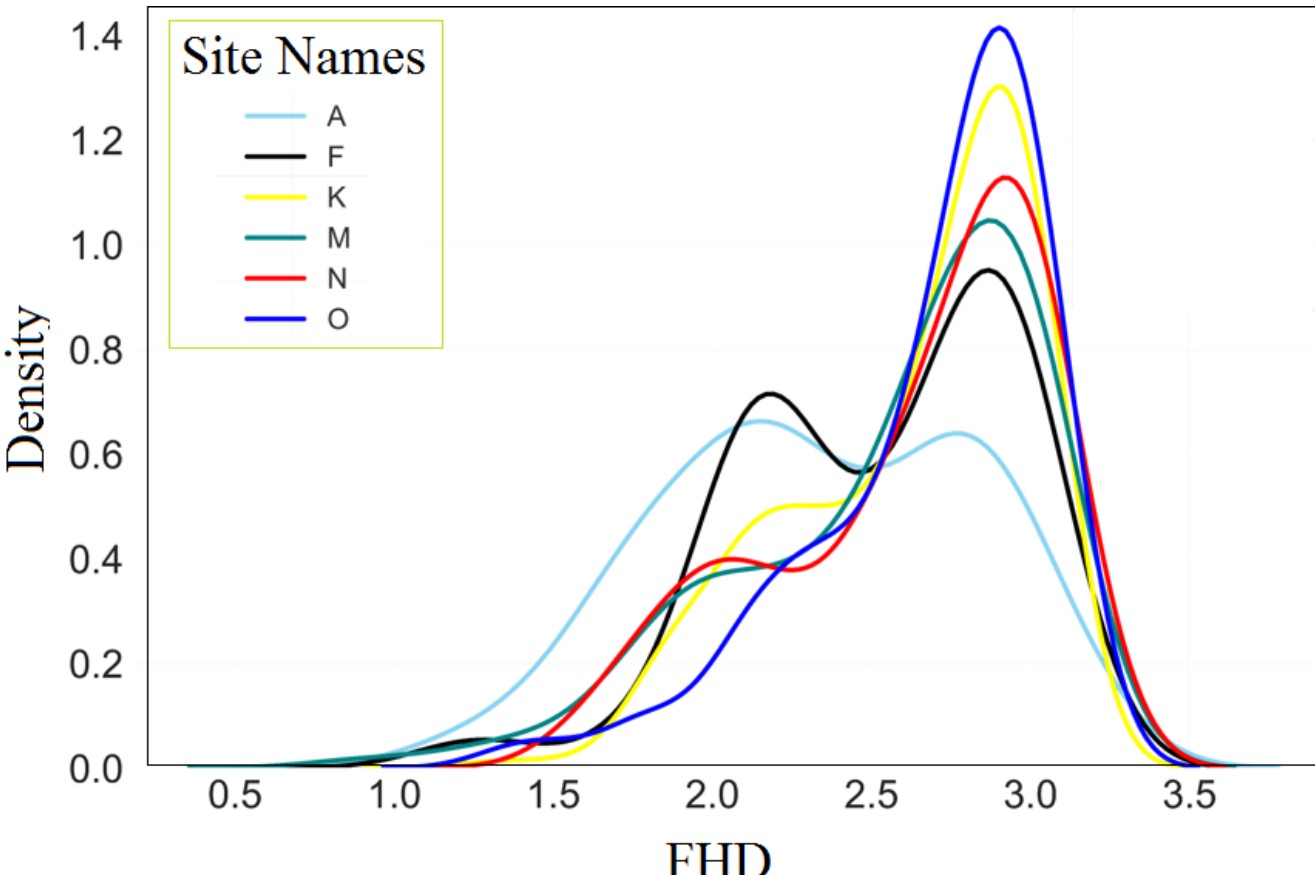

**Figure B3: Foliage Height Diversity (FHD) distribution for each site.**

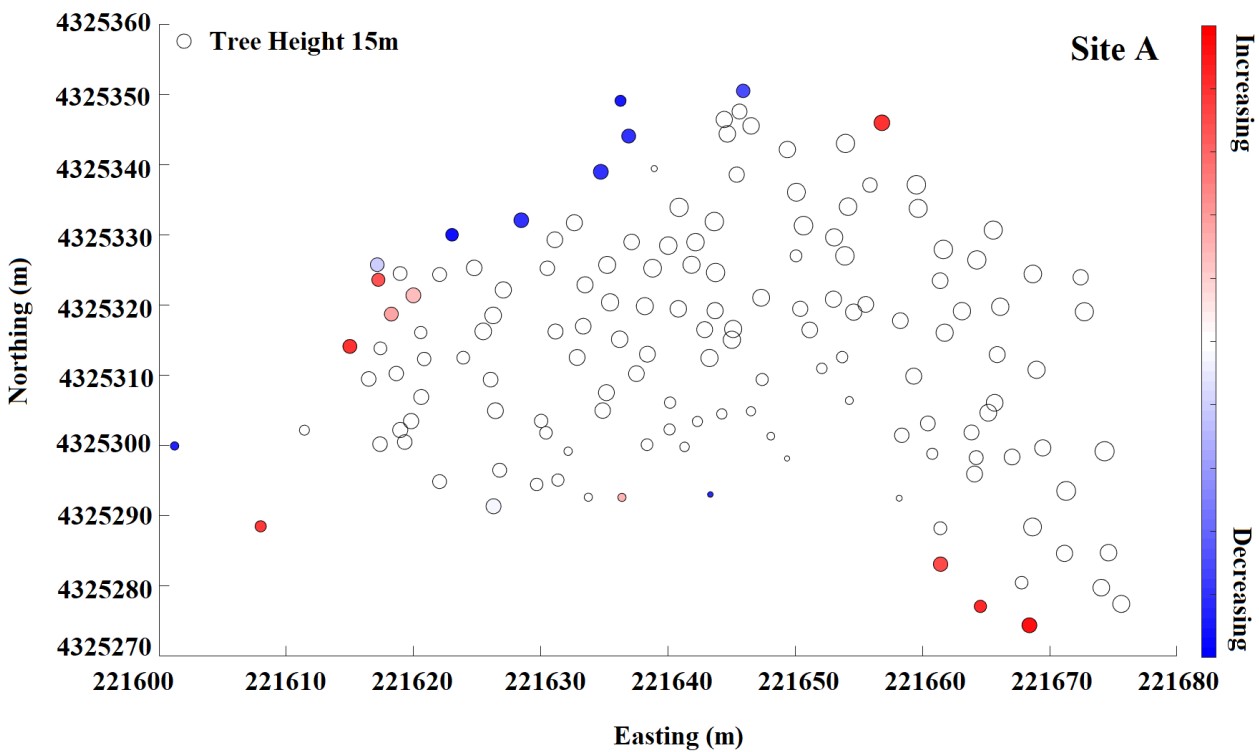

**Figure B4: Snow depth change within a 10 m buffer from the edge of a tree at site A. Red color indicates snow depth increases moving from the tree edge toward the open and blue indicates snow depth decreases. Increasing and decreasing patterns are shown for individual trees at each site with adequate snow coverage.**


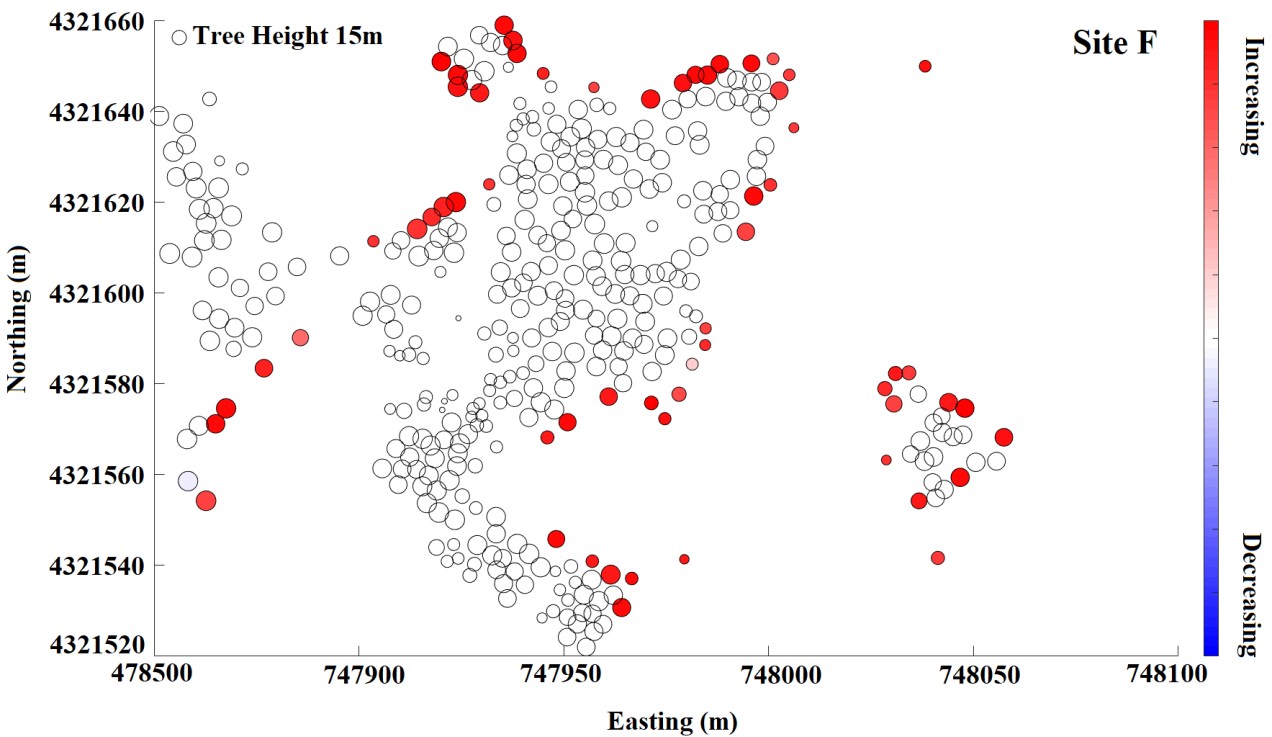

**Figure B5: Snow depth change within a 10 m buffer from the edge of a tree at site F. Red color indicates snow depth increases moving from the tree edge toward the open and blue indicates snow depth decreases. Increasing and decreasing patterns are shown for individual trees at each site with adequate snow coverage.**

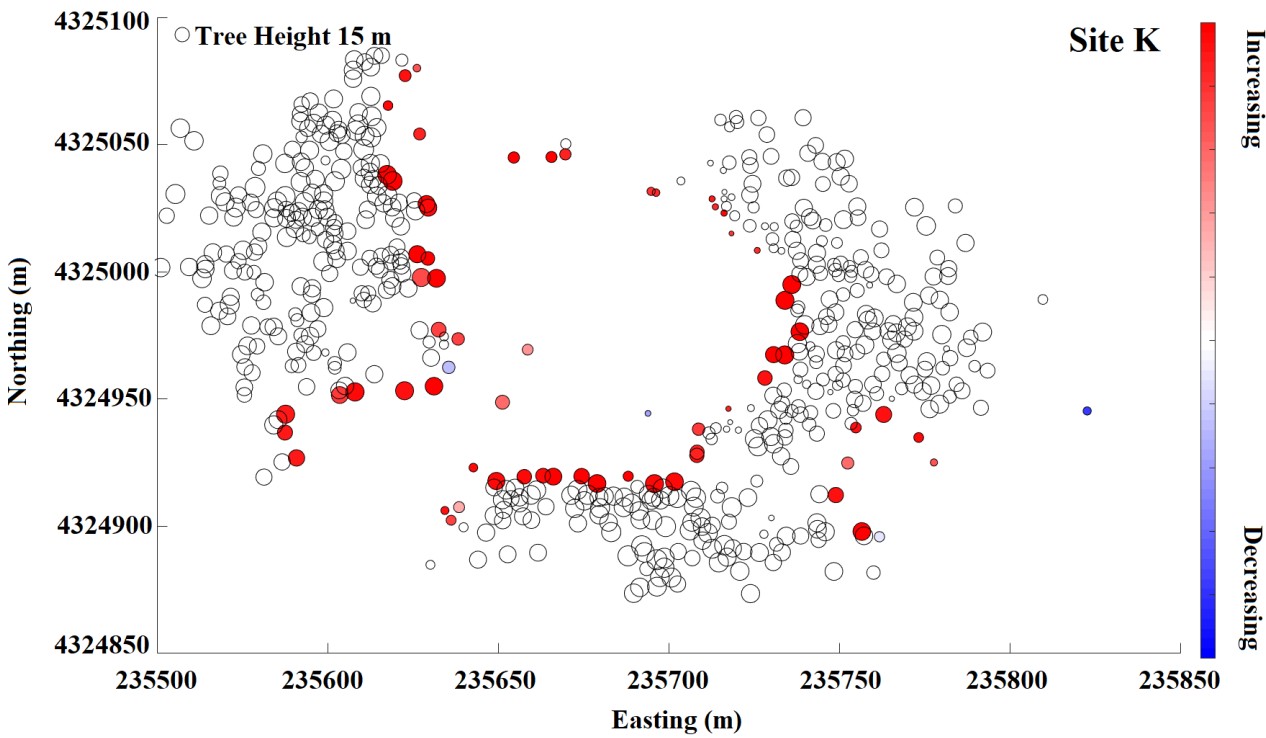


**Figure B6: Snow depth change within a 10 m buffer from the edge of a tree at site K. Red color indicates snow depth increases moving from the tree edge toward the open and blue indicates snow depth decreases. Increasing and decreasing patterns are shown for individual trees at each site with adequate snow coverage.**

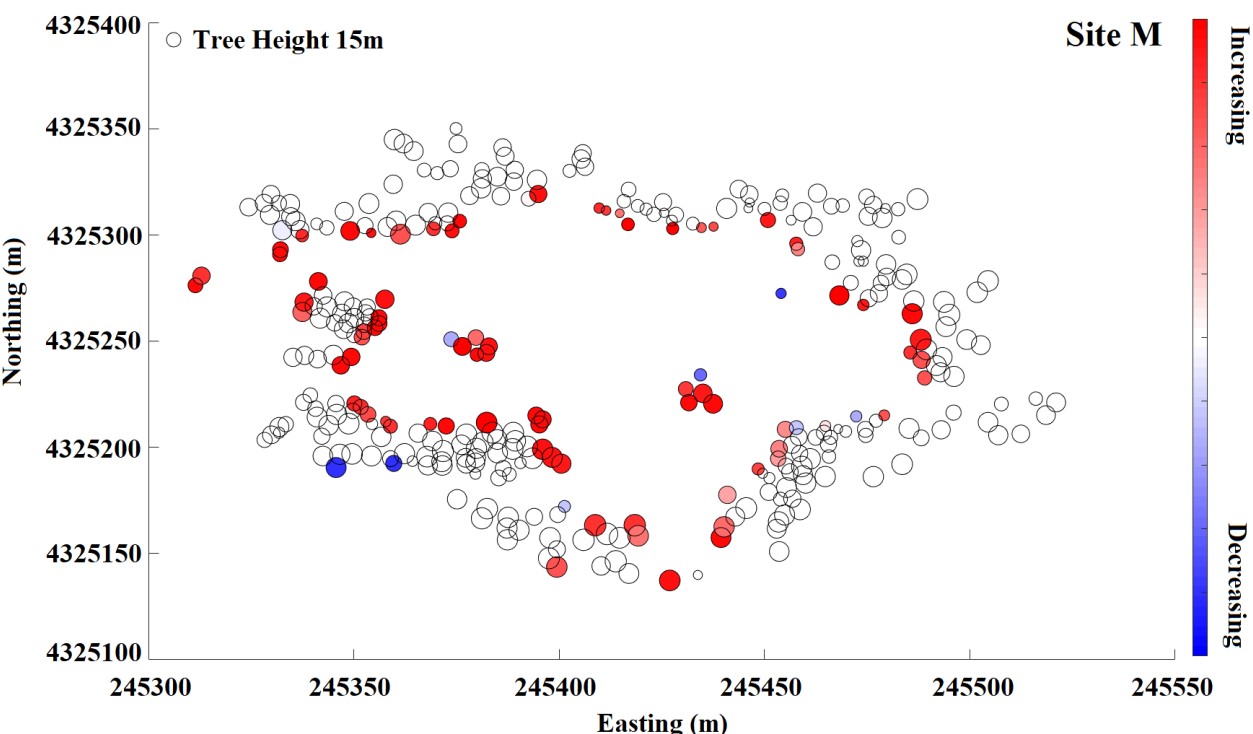

**Figure B7: Snow depth change within a 10 m buffer from the edge of a tree at site M. Red color indicates snow depth increases moving from the tree edge toward the open and blue indicates snow depth decreases. Increasing and decreasing patterns are shown for individual trees at each site with adequate snow coverage.**

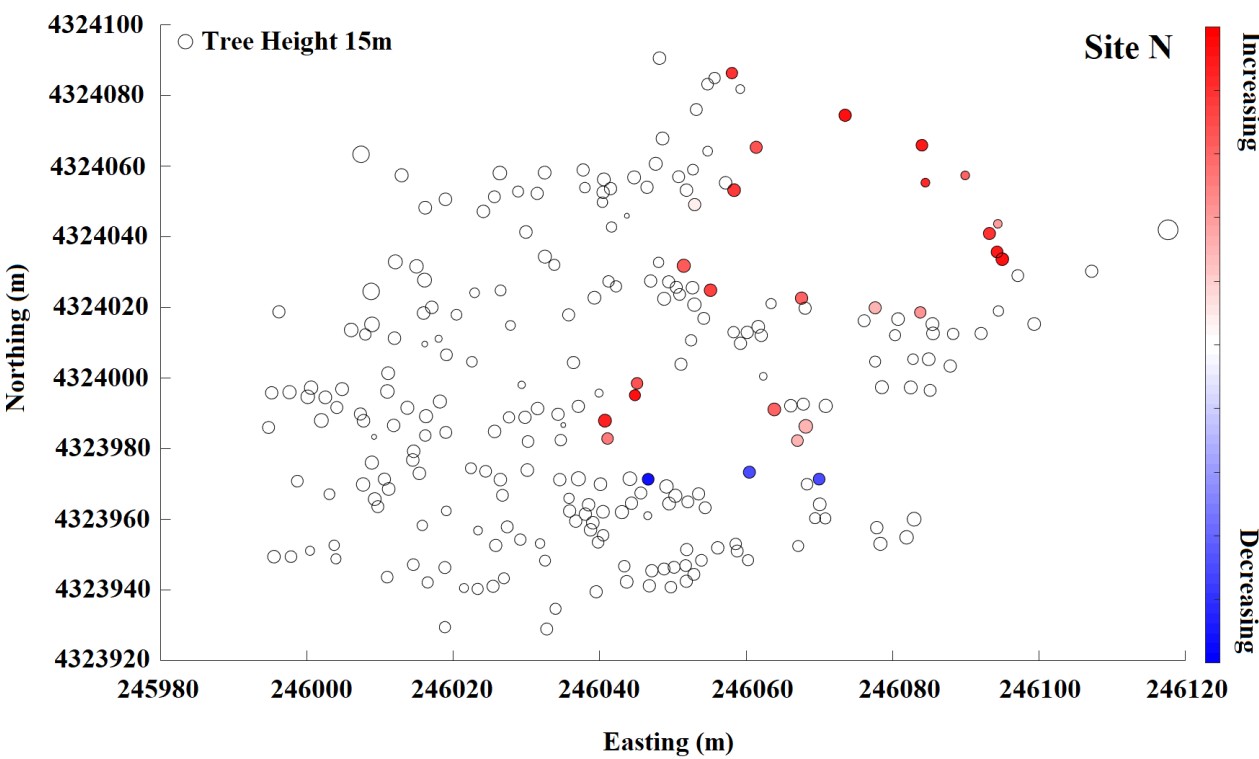

**Figure B8: Snow depth change within a 10 m buffer from the edge of a tree at site N. Red color indicates snow depth increases moving from the tree edge toward the open and blue indicates snow depth decreases. Increasing and decreasing patterns are shown for individual trees at each site with adequate snow coverage.**


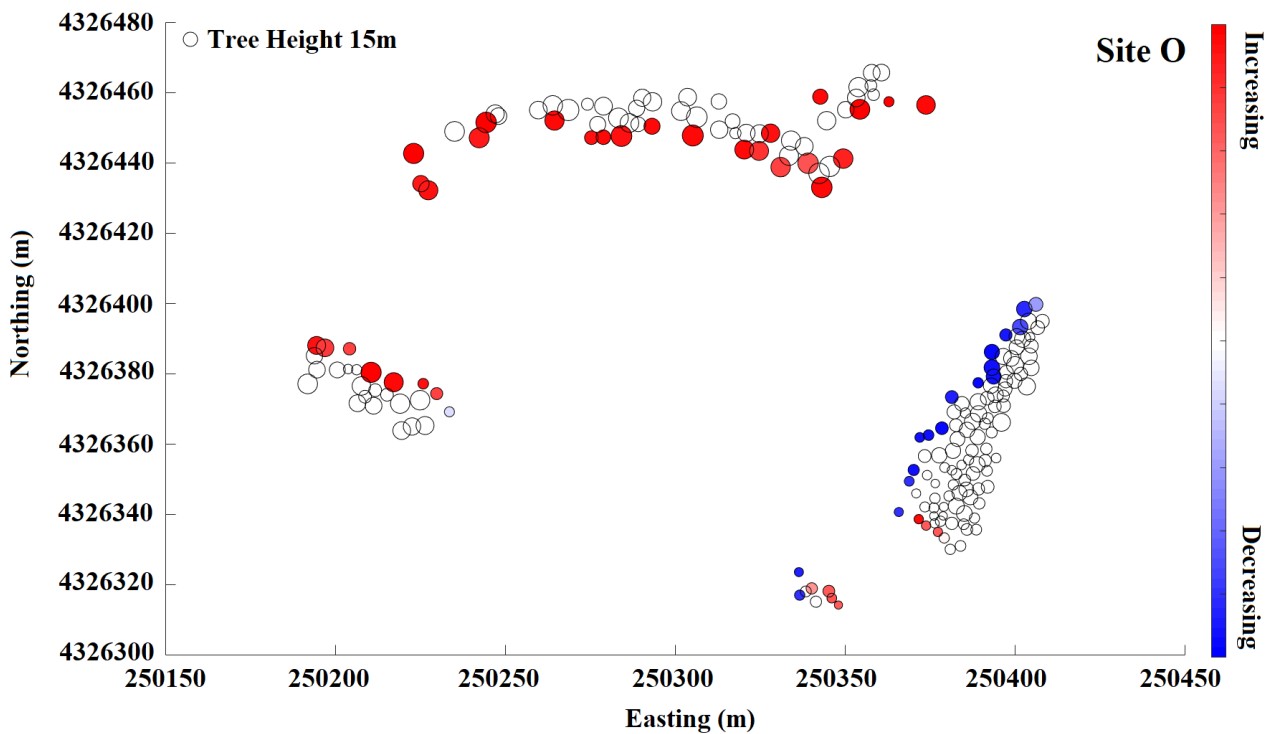

**Figure B9: Snow depth change within a 10 m buffer from the edge of a tree at site O. Red color indicates snow depth increases moving from the tree edge toward the open and blue indicates snow depth decreases. Increasing and decreasing patterns are shown for individual trees at each site with adequate snow coverage.**
