# Peer review of "Tree canopy and snow depth relationships at fine scales with terrestrial laser scanning"

_The Cryosphere, 2020_

## Referee Comment (RC1) · Phillip Harder (Referee) · 27 Nov 2020

Review of Hojatimalekshah et al., "Tree canopy and snow depth relationships at fine scales with terrestrial laser scanning"

Relating tree structure to snow depth dynamics is a very important research area with cryospheric and hydrologic implications. It is also a very challenging interaction to quantify and this study uses a Terrestrial Laser Scanning (TLS) dataset to examine such dynamics at a selection of site's from Grand Mesa in 2017. Interesting relationships are revealed in terms canopy and topographic controls. This paper would make a better contribution with a number of changes, generally around clarifying the storyline and focusing the results presented, and therefore I would recommend major revisions

at this point prior.

My review begins with my major comments and followed by some specific comments. At this point I will not make technical comments.

Literature context in Introduction:

Overall I enjoyed the introduction and how it established the context for the rest of the paper. This would further be strengthened by a concise description of the physical processes (interception/sublimation and blowing snow differences between forest and clearing) to help interpret the findings later. As well there is a focus on airborne lidar and terrestrial lidar as the only tools by which to quantify snow-forest interactions. There is a large chunk of recent literature that has been starting to employ drones (with structure from motion- many including Buhler 2016, Harder et al., 2016; Vander Jagt et al., 2015; De Michele et al., 2016, Walker et al., 2020. SfM does have trouble with dense vegetation but would still be a method in the sites with more sparse tree cover. More recently lidar on drones for snow depth (Harder et al., 2020 and Jacobs et al., 2020) has been demonstrated to deal with vegetation better than ALS). Drone scales bridge TLS to airborne scales so is relevant in this discussion and should not be ignored. For full disclosure I do work with UAV's and snow and this is not reviewer coerced citation for selfish purposes– UAV scale and capabilities can and do work on these same scales/problems and should not be ignored.

TLS for forests:

The path of photons from a TLS are oblique and so point densities quickly diminish as one moves further away the exposed side of a tree/tree stand and point densities diminish from the occlusion in vegetation. This is evident from the footprint of the analysis and the snow depth maps reported in the appendix. This is not discussed as a limitation anywhere that I can recall in this paper. TLS is not a new method to capture edge of forest snow interactions and am not discounting previous work. I guess that I would like to see a discussion/acknowledgment of this. Especially when

we are asked to consider the relationship with respect to side of tree. Won't this dataset be biased to capture interactions in greater detail on the exposure side rather than the obscured size. Will this bias the results/ relationships? How do the vegetation metrics respond to a TLS scan which captures one side of a tree better than the other? Also is there a threshold to determine the extent of the analysis and how is that determined? Considering these ideas should this be paper/title be rather reframed as an analysis of snow depth-tree relationships for sparse/gappy areas. As it is presented it would seem that these findings should be pertinent to all forests but from the snow depth maps we can see that extent of the datasets are very much limited to isolated/sparse trees and edge of forest areas?

Results:

There are a lot of results represented in various figures and tables that is on the overwhelming side of things. Can this be significantly pared down to the most important findings or the conversion of some large tables into figures (Table a1)? With 32 possible descriptor variables that have variable levels of description and no hypotheses presented it implies more of a fishing expedition which is not ideal. Are there metrics that have been used previously besides tree height? Could you focus on metrics that are a bit more common – ie LAI or sky view factor are some that come to mind- so that these findings could have broader applicability. FHD is not a commonly used metric yet. In a revision could this be flipped and some specific hypotheses be tested? Would help to focus things. Need more help with interpretation of Figure B8, B9 (text way too small), 7 (why are the positive and negative scales split). Figure b1 should be more prominent – very useful for interpretation of all of this data. Could the results and discussion be grounded more strongly in the physical process descriptions – correlations themselves in specific situations can be hard to parse.

Specific comments:

I struggled to understand what was being communicated in the paragraph in lines 63-

68. What does "observed the high contribution of storms in defining the snow accumulation pattern" mean in this context? Can it be clearer what proper scales are and how that is related to the controlling processes?

Line 120: Can you elaborate on "thus we reclassified these points manually using the software TerraScan". Not reproducible without knowing what the manual procedures implemented were.

Section 2.3.2: perhaps a graphic could be used to explain M3C2?

Line 130-131: Can you elaborate on how transition zones were classified?

Line 155: "contain a minimum on 10 snow pixels." The tree polygons are variable in size yes? Perhaps this would be more robust if a percentage of area needed snow pixels? Is this how low density snow depth areas are removed from the analysis?

Section 2.3.6: What tool/software was used?

Section 3.6: Did it result in deeper snow? You have the data to test this, correct?

Line 250: how representative are findings that are limited to one side of a dense stand of trees to make a comment on tree-snow directional dynamics. A limitation with TLS. Can findings be modified to account for the bias?

Site naming: Could the sites be named in a way that can simply convey some of their main features. Lettering doesn't convey much and would make tracking the relationships a bit easier.

Observation temporal extent: Can it be emphasized more clearly that these were single measurements (not multi-temporal) and primarily reflect snow accumulations processes. Things will obviously be different if needing to account for snowmelt dynamics.

I appreciate the challenges in relating snow depth and vegetation metrics at fine scales with real world data and look forward to seeing your response.

Phillip Harder

References:

Bühler, Y., Adams, M. S., Bösch, R., and Stoffel, A.: Mapping snow depth in alpine terrain with unmanned aerial systems (UASs): potential and limitations, The Cryosphere, 10, 1075– 1088, https://doi.org/10.5194/tc-10-1075-2016, 2016

De Michele, C., Avanzi, F., Passoni, D., Barzaghi, R., Pinto, L., Dosso, P., Ghezzi, A., Gianatti, R., and Della Vedova, G.: Using a fixed-wing UAS to map snow depth distribution: an evaluation at peak accumulation, The Cryosphere, 10, 511–522, https://doi.org/10.5194/tc-10-511-2016, 2016.

Harder, P., Schirmer, M., Pomeroy, J., and Helgason, W.: Accuracy of snow depth estimation in mountain and prairie environments by an unmanned aerial vehicle, The Cryosphere, 10, 2559–2571, https://doi.org/10.5194/tc-10-2559-2016, 2016

Harder P., Pomeroy J.W. And Helgason W.D. (2020) Improving sub-canopy snow depth mapping with unmanned aerial vehicles: lidar versus structure-from-motion techniques.The Cryosphere: 14, pp. 1919-1935 DOI: 10.5194/tc-14-1919-2020

Jacobs, J. M., Hunsaker, A. G., Sullivan, F. B., Palace, M., Burakowski, E. A., Herrick, C., and Cho, E.: Shallow snow depth mapping with unmanned aerial systems lidar observations: A case study in Durham, New Hampshire, United States, The Cryosphere Discuss., https://doi.org/10.5194/tc-2020-37, in review, 2020.

Vander Jagt, B., Lucieer, A., Wallace, L., Turner, M., and Durand, D.: Snow Depth Retrieval with UAS Using Photogrammetric Techniques, Geosciences, 5, 264–285, https://doi.org/10.3390/geosciences5030264, 2015.

Walker B, Wilcox E, and Marsh P. ACCURACY ASSESSMENT OF LATE WINTER SNOW DEPTH MAPPING FOR TUNDRA ENVIRONMENTS USING STRUCTURE-FROM-MOTION PHOTOGRAMMETRY. Arctic Science. 0(ja): -. https://doi.org/10.1139/AS-2020-0006

---

## Referee Comment (RC2) · Anonymous Referee #2 · 28 Nov 2020

General comments

In this work, the authors focus on quantifying the roles of tree structure (and wind and topographic characteristics such as slope and aspect) in controlling snow depth variations using terrestrial laser scanning (TLS). The authors found that vegetation structural metrics (foliage height diversity) and wind are highly influential on spatial variability of snow depth. They also highlighted that windward slopes have greater impact on snow accumulation than vegetation features.

Overall, I enjoyed reading this manuscript. The authors identified interesting research questions and attempted to leverage the strength of the TLS data to fill in the scientific gaps. However, the current version of this work needs to be improved upon before publication in The Cryosphere is warranted. Major concerns are given below. I would

recommend that this paper be returned for major revisions and specifically request inclusions of additional analysis with appropriate interpretations and reorganizing the structures of the manuscript for the Cryosphere community.

Major comments:

1. A major concern is that the methodology used to derive the conclusion in this study is entirely based on correlation coefficient in a linear regression without investigating inter-dependency among the physical variables. I do not think if the linear correlation-based approach is enough to identify and to conclude the relationship between snow depth and the tree variables. For example, the snow depth with the distance from the canopy edge is not linear (see Figure 2 in Hardy Albert, 1995). A potential approach I would suggest can be regression tree or multivariable analysis (e.g. multiple linear/logistic regression analysis with the standardized coefficient) to quantify relative contribution of the vegetation metrics along with wind, topographic features (Molotch et al., 2005; Schneider et al., 2020). Also, it would be beneficial to provide variation in snow depth, key vegetation characteristics, and elevation, etc along representative transects for the sites. Please refer to Sturm Fig.2 in Sturm et al. (2001).

- Molotch, N. P., Colee, M. T., Bales, R. C. Dozier, J. 2005. Estimating the spatial distribution of snow water equivalent in an alpine basin using binary regression tree models: the impact of digital elevation data and independent variable selection. Hydrological Processes 19 (7), 1459–1479. doi:10. 1002/hyp.5586

- Schneider, D., Molotch, N. P., Deems, J. S., Painter, T. H. (2020). Analysis of topographic controls on depletion curves derived from airborne lidar snow depth data. Hydrology Research.

- Hardy, J. P., Albert, M. R. (1995). Snowâ Ăˇ Rinduced thermal variations around a single conifer tree. Hydrological processes, 9(8), 923-933.

- Sturm, M., Holmgren, J., McFadden, J. P., Liston, G. E., Chapin III, F. S., Racine,

C. H. (2001). Snow–shrub interactions in Arctic tundra: a hypothesis with climatic implications. Journal of Climate, 14(3), 336-344.

2. Regarding the comment above, another concern is the interpretation of the correlation coefficient values from the linear regressions in the result sections. For example, the authors state that "slope explained 44

3. I think the TLS data's reliability should be verified. How accuracy is TLS-based snow depth, especially under the tree canopy? Many previous studies found that there were issues in ALS, TLS, structure-from-motion photogrammetry (SfM) with observation gaps in forested regions. The return density under or near canopy can be extremely low that may not be adequate to observe spatial variations of snow depth. In Figure B1, the snow depth maps the authors provided seem to be very limited to areas near forest and under the trees. Thus, I would recommend that the author quantify the accuracy of the snow depth measurements especially under/near canopy. Have you seen comparison results with independent snow depth measurements? I know the validation work is out of scope in this study, but it would be helpful for readers to be able to have a sense of how accuracy the TSL technique is, particularly in these sites. I believe there are various available ground-based (or other techniques) snow depth measurements because this work was part of the NASA-led SnowEx 2017 campaign. If they are not available, the authors should provide at least general uncertainties in TLS-based snow depth from previous findings, particularly under/near canopy.

4. There are a few comments in terms of structure of the paper. (1) Given the three research questions in L74-78, it would be best to either rearrange the results (discussion) to better address the three questions or increase the number of questions to better reflect the structure that the results are provided. Too many subsections exist in the result section. I would recommend combining the subsections based upon the questions. Also please consider to combine "Results" and "Discussion" (because some descriptions in the both sections are duplicated). (2) I would strongly suggest reorganizing the figures and tables in the main body and the supplementary. I think

some figures and tables (e.g. Figure 4 and Table 2) in the main body would be better to be included in the supplementary. Similarly, some figures and tables in the supplementary should be moved into the main body (Figure B1). (3) Some sections should be renamed and relocated. For example, "2.1 Study area" should not be under "2. Method". "2.2 TLS" and "2.3 TLS Data Processing" should be combined which may be under "Data and preprocessing". Also, I think contents in some subsections are too short to comprise an individual subsection (e.g. Section 3.6 and 3.7). It would be good to combine similar subsections into one.

Specific comments

L41 Please add citations

L58 Please provide a range of the snow depth quantitively (e.g. snow depth > XX mm) with general forest information (e.g. dominant types).

L63-64 Would you check the reference again? Schirmer et al. (2011) do not provide the relationship between vegetation or canopy characteristics and wind effect and snow depth variations.

L200 Can you quantify what "mid-to-high correlation" mean? Also I would recommend providing correlation matrixes between vegetation metrics and snow depth for each site to identify intercorrelation among the vegetation metrics.

L203 two distributions -> two peaks of the FHD distribution

L204 0.35-0.75 -> -0.35 to -0.75; throughout the manuscript

L222 Remove "we propose"

L304-205 This is identical to the sentence above. Rephrase "more evenly spaced along an individual tree"

L273-274 Rephrase the sentence.

L292-294 This should be in data or methodology sections.

Table 1 In Median Absolute Deviation, what is the constant number "1.4826"?

Table A2 Please add the units. I think boxplots would be more suitable to present the dataset. For example, 3x6 boxplots with three different colors for canopy, transition, and open areas.

Figure 4 It would be fine to move into Appendix.

Figure B9 In the figure, site O has random distribution. But in the caption "Only site N has a random distribution pattern of trees". Please double check. And I do not think the six distributions are needed in this figure even in supplementary info – all distributions are the same. A table including nearest neighbor values only would be more appropriate than the figure.

---

## Author Comment (AC1) · 18 Jan 2021

Thank you for all the helpful comments! Please see our responses (in italics) with intended revisions if this manuscript moves to the next stage.

RC1: Review of Hojatimalekshah et al., "Tree canopy and snow depth relationships at fine scales with terrestrial laser scanning"

Relating tree structure to snow depth dynamics is a very important research area with cryospheric and hydrologic implications. It is also a very challenging interaction to quantify and this study uses a Terrestrial Laser Scanning (TLS) dataset to examine such dynamics at a selection of site's from Grand Mesa in 2017. Interesting relationships are revealed in terms canopy and topographic controls. This paper would make a

better contribution with a number of changes, generally around clarifying the storyline and focusing the results presented, and therefore I would recommend major revisions at this point prior.

My review begins with my major comments and followed by some specific comments.

At this point, I will not make technical comments.

Literature context in Introduction:

Overall, I enjoyed the introduction and how it established the context for the rest of the paper.

1. This would further be strengthened by a concise description of the physical processes (interception/sublimation and blowing snow differences between forest and clearing) to help interpret the findings later.

Thank you for the suggestion. We will add details to the Introduction re: physical processes between forest and non-forested areas.

2. As well, there is a focus on airborne lidar and terrestrial lidar as the only tools by which to quantify snow-forest interactions. There is a large chunk of recent literature that has been starting to employ drones (with structure from motion- many including Buhler 2016, Harder et al., 2016; Vander Jagt et al., 2015; De Michele et al., 2016, Walker et al., 2020. SfM does have trouble with dense vegetation but would still be a method in the sites with more sparse tree cover. More recently, lidar on drones for snow depth (Harder et al., 2020 and Jacobs et al., 2020) has been demonstrated to deal with vegetation better than ALS). Drone scales bridge TLS to airborne scales so is relevant in this discussion and should not be ignored. For full disclosure I do work with UAV's and snow and this is not reviewer coerced citation for selfish purposes– UAV scale and capabilities can and do work on these same scales/problems and should not be ignored. This is a good point and we appreciate the references! We will add a sentence or two to both the Introduction and Discussion to include recent studies and

demonstrate the value of UAS in snow-veg studies.

TLS for forests:

1. The path of photons from a TLS are oblique and so point densities quickly diminish as one moves further away the exposed side of a tree/tree stand and point densities diminish from the occlusion in vegetation. This is evident from the footprint of the analysis and the snow depth maps reported in the appendix. This is not discussed as a limitation anywhere that I can recall in this paper. TLS is not a new method to capture edge of forest snow interactions and am not discounting previous work. I guess that I would like to see a discussion/acknowledgment of this. Especially when we are asked to consider the relationship with respect to side of tree. Won't this dataset be biased to capture interactions in greater detail on the exposure side rather than the obscured size. Will this bias the results/ relationships?

We utilized multiple scan positions to avoid occlusion of the trees we used for analysis; however as you point out, there are some instances where occlusion occurred. We avoided this for our canopy edge analyses by only using tree polygons with sufficient coverage (see Section 2.3.4). We agree that it is important to add a statement about occlusion in our Discussion as a challenge of TLS, and we could also discuss this in the context of the complementary nature of UAS and airborne lidar.

2. How do the vegetation metrics respond to a TLS scan which captures one side of a tree better than the other? Also is there a threshold to determine the extent of the analysis and how is that determined?

These are good questions and we'd like to clarify. We did not test for vegetation metrics based on a subsampling approach (on one side of a tree versus the other). To avoid unbalanced samples from different sides of a tree we used the whole tree to compute the related metrics. In addition, each tree was scanned from at least three different scan positions. However, there is still uncertainty about trees being evenly scanned, specifically within dense canopies which we can mention in the Discussion. Specific to

our directional analysis, we used tree heights (only) to correlate to snow depth per each direction. We do not expect tree height to be affected by any sampling or occlusion issues. We will clarify this in the Discussion.

3. Considering these ideas should this be paper/title be rather reframed as an analysis of snow depth-tree relationships for sparse/gappy areas. As it is presented it would seem that these findings should be pertinent to all forests but from the snow depth maps we can see that extent of the datasets are very much limited to isolated/sparse trees and edge of forest areas?

We understand what you mean given that our images show plots that capture edges. Our analyses use individual trees in rather dense canopies (please see Fig B9, which will likely become a table in the revision) and we utilize edges of canopies - we will ensure that this is clear in a revision. We will also ensure that our Discussion indicates that our interpretations cannot be used for general assumptions / forests with different characteristics of this study area. For these reasons we feel that our title is appropriate as is.

Results:

1. There are a lot of results represented in various figures and tables that is on the overwhelming side of things. Can this be significantly pared down to the most important findings or the conversion of some large tables into figures (Table a1)? With 32 possible descriptor variables that have variable levels of description and no hypotheses presented it implies more of a fishing expedition which is not ideal. Are there metrics that have been used previously besides tree height? Could you focus on metrics that are a bit more common – ie LAI or sky view factor are some that come to mind- so that these findings could have broader applicability. FHD is not a commonly used metric yet. In a revision could this be flipped and some specific hypotheses be tested? Would help to focus things.

Yes, we will review the large tables (1 and 2) and attempt to put appropriate information
in a figure. The use of the different vegetation variables are in response to question 1 of the manuscript. FHD is used in ecology and while it may not be commonly used in cryosphere studies, we hope to integrate these fields. In our revision we can provide more context for the use of FHD.

Yes, we can add hypothesis in lieu of, or in addition, to our questions if need be.

2. Need more help with interpretation of Figure B8, B9 (text way too small), 7 (why are the positive and negative scales split).

Thank you for the suggestion. We can provide more detail in the captions in Fig B8. We will change Figure B9 into a table for clarity. We split the negative and positive scales for easier interpretation. However, we will merge those to one.

3. Figure b1 should be more prominent – very useful for interpretation of all of this data.

Thanks for the suggestion. We will move it to the manuscript.

4. Could the results and discussion be grounded more strongly in the physical process descriptions – correlations themselves in specific situations can be hard to parse.

Yes, we will revise the Discussion to ground our observations within the context of physical processes.

Specific comments:

1. I struggled to understand what was being communicated in the paragraph in lines 63-68. What does "observed the high contribution of storms in defining the snow accumulation pattern" mean in this context? Can it be clearer what proper scales are and how that is related to the controlling processes?

Thanks. We will clarify these sentences - we were trying to emphasize the need for fine-scale TLS data.

2. Line 120: Can you elaborate on "thus we reclassified these points manually using the software TerraScan". Not reproducible without knowing what the manual procedures implemented were.

We will clarify in the text that manual classification included visually separating snow under the trees from tree trunks. Yes, we agree this isn't completely reproducible but adding these details will help.

3. Section 2.3.2: perhaps a graphic could be used to explain M3C2?

Yes, we can do this and add to the Supplementary material.

4. Line 130-131: Can you elaborate on how transition zones were classified?

Yes, we will clarify that the transition zone consisted of defining a 10 m outer buffer beyond each tree polygon in the direction of the open.

5. Line 155: "contain a minimum on 10 snow pixels." The tree polygons are variable in size yes? Perhaps this would be more robust if a percentage of area needed snow pixels? Is this how low density snow depth areas are removed from the analysis?

Thank you for pointing this out. We will consider this in our analysis.

6. Section 2.3.6: What tool/software was used?

We will clarify that we used ArcMap.

7. Section 3.6: Did it result in deeper snow? You have the data to test this, correct?

Yes, the mean snow depth is presented in Table A2.

8. Line 250: how representative are findings that are limited to one side of a dense stand of trees to make a comment on tree-snow directional dynamics. A limitation with TLS. Can findings be modified to account for the bias?

Thanks for pointing this out - we need to clarify that our comments are specific to site A and refer the reader to Fig 6b, and reiterate the sampling limitation of TLS.

Site naming:

1. Could the sites be named in a way that can simply convey some of their main features. Lettering doesn't convey much and would make tracking the relationships a bit easier.

We agree though we would prefer to keep the lettering as is, as they correspond to the NASA SnowEx project naming convention.

2. Observation temporal extent: Can it be emphasized more clearly that these were single measurements (not multi-temporal) and primarily reflect snow accumulations processes. Things will obviously be different if needing to account for snowmelt dynamics.

Yes, thank you and we will more clearly point this out in the methods and discussion.

I appreciate the challenges in relating snow depth and vegetation metrics at fine scales with real world data and look forward to seeing your response.

Thank you and we appreciate these helpful comments and references!

References:

Bühler, Y., Adams, M. S., Bösch, R., and Stoffel, A.: Mapping snow depth in alpine terrain with unmanned aerial systems (UASs): potential and limitations, The Cryosphere, 10, 1075– 1088, https://doi.org/10.5194/tc-10-1075-2016, 2016

De Michele, C., Avanzi, F., Passoni, D., Barzaghi, R., Pinto, L., Dosso, P., Ghezzi, A., Gianatti, R., and Della Vedova, G.: Using a fixed-wing UAS to map snow depth distribution: an evaluation at peak accumulation, The Cryosphere, 10, 511–522, https://doi.org/10.5194/tc-10-511-2016, 2016.

Harder, P., Schirmer, M., Pomeroy, J., and Helgason, W.: Accuracy of snow depth estimation in mountain and prairie environments by an unmanned aerial vehicle, The Cryosphere, 10, 2559–2571, https://doi.org/10.5194/tc-10-2559-2016, 2016

Harder P., Pomeroy J.W. And Helgason W.D. (2020) Improving sub-canopy snow depth mapping with unmanned aerial vehicles: lidar versus structure-from-motion techniques. The Cryosphere: 14, pp. 1919-1935 DOI: 10.5194/tc-14-1919-2020

Jacobs, J. M., Hunsaker, A. G., Sullivan, F. B., Palace, M., Burakowski, E. A., Herrick, C., and Cho, E.: Shallow snow depth mapping with unmanned aerial systems lidar observations: A case study in Durham, New Hampshire, United States, The Cryosphere Discuss., https://doi.org/10.5194/tc-2020-37, in review, 2020.

Vander Jagt, B., Lucieer, A., Wallace, L., Turner, M., and Durand, D.: Snow Depth Retrieval with UAS Using Photogrammetric Techniques, Geosciences, 5, 264–285, https://doi.org/10.3390/geosciences5030264, 2015.

Walker B, Wilcox E, and Marsh P. ACCURACY ASSESSMENT OF LATE WINTER SNOW DEPTH MAPPING FOR TUNDRA ENVIRONMENTS USING STRUCTURE-FROM-MOTION PHOTOGRAMMETRY. Arctic Science. 0(ja): - .https://doi.org/10.1139/AS-2020-0006

---

## Author Comment (AC2) · 18 Jan 2021

Thank you for all the helpful comments! Please see our responses (in italics) with intended revisions if this manuscript moves to the next stage.

RC2: General comments

In this work, the authors focus on quantifying the roles of tree structure (and wind and topographic characteristics such as slope and aspect) in controlling snow depth variations using terrestrial laser scanning (TLS). The authors found that vegetation structural metrics (foliage height diversity) and wind are highly influential on spatial variability of snow depth. They also highlighted that windward slopes have greater impact on snow accumulation than vegetation features.

[Figure]

Overall, I enjoyed reading this manuscript. The authors identified interesting research questions and attempted to leverage the strength of the TLS data to fill in the scientific gaps. However, the current version of this work needs to be improved upon before publication in The Cryosphere is warranted. Major concerns are given below. I would recommend that this paper be returned for major revisions and specifically request inclusions of additional analysis with appropriate interpretations and reorganizing the structures of the manuscript for the Cryosphere community.

Major comments:

1. A major concern is that the methodology used to derive the conclusion in this study is entirely based on correlation coefficient in a linear regression without investigating inter-dependency among the physical variables. I do not think if the linear correlation based approach is enough to identify and to conclude the relationship between snow depth and the tree variables. For example, the snow depth with the distance from the canopy edge is not linear (see Figure 2 in Hardy Albert, 1995). A potential approach I would suggest can be regression tree or multivariable analysis (e.g. multiple linear/logistic regression analysis with the standardized coefficient) to quantify relative contribution of the vegetation metrics along with wind, topographic features (Molotch et al., 2005; Schneider et al., 2020). Also, it would be beneficial to provide variation in snow depth, key vegetation characteristics, and elevation, etc along representative transects for the sites. Please refer to Sturm Fig.2 in Sturm et al. (2001).

Thank you, we appreciate the comments and references. We will investigate the multicollinearity of the variables, and the use of a multiple linear regression, and make subsequent changes in the results and interpretations. While our analyses were performed at individual trees, we can investigate if the data will support representative transects.

Âů Molotch, N. P., Colee, M. T., Bales, R. C. Dozier, J. 2005. Estimating the spatial distribution of snow water equivalent in an alpine basin using binary regression tree

models: the impact of digital elevation data and independent variable selection. Hydrological Processes 19 (7), 1459–1479. doi:10. 1002/hyp.5586

Âů Schneider, D., Molotch, N. P., Deems, J. S., Painter, T. H. (2020). Analysis of topographic controls on depletion curves derived from airborne lidar snow depth data.Hydrology Research.

Âů Hardy, J. P., Albert, M. R. (1995). SnowâËŸARËĞ induced thermal variations around a single conifer tree. Hydrological processes, 9(8), 923-933.

Âů Sturm, M., Holmgren, J., McFadden, J. P., Liston, G. E., Chapin III, F. S., Racine, C. H. (2001). Snow–shrub interactions in Arctic tundra: a hypothesis with climatic implications. Journal of Climate, 14(3), 336-344.

2. Regarding the comment above, another concern is the interpretation of the correlation coefficient values from the linear regressions in the result sections. For example, the authors state that "slope explained 44

While this comment was cut off, we think the reviewer is concerned about 'slope explained 44% of the variance' and how this is interpreted. This will be clarified in the process of responding to the above comment #1.

3. I think the TLS data's reliability should be verified. How accuracy is TLS-based snow depth, especially under the tree canopy? Many previous studies found that there were issues in ALS, TLS, structure-from-motion photogrammetry (SfM) with observation gaps in forested regions. The return density under or near canopy can be extremely low that may not be adequate to observe spatial variations of snow depth. In Figure B1, the snow depth maps the authors provided seem to be very limited to areas near forest and under the trees. Thus, I would recommend that the author quantify the accuracy of the snow depth measurements especially under/near canopy. Have you seen comparison results with independent snow depth measurements? I know the validation work is out of scope in this study, but it would be helpful for readers to be

able to have a sense of how accuracy the TSL technique is, particularly in these sites. I believe there are various available ground-based (or other techniques) snow depth measurements because this work was part of the NASA-led SnowEx 2017 campaign. If they are not available, the authors should provide at least general uncertainties in TLS-based snow depth from previous findings, particularly under/near canopy.

Thank you for the comment. A previous paper (Currier et al., 2019) demonstrates the relative accuracy of the TLS to airborne lidar (ALS), which is likely a better measure than using the snow depth measurements that were not collected at all sites and/or same day. Currier et al., indicate that the median snow depth difference between ALS and TLS at sites A and K was less than 5 cm. In addition, ALS snow depth comparisons with field transects indicated that the median values for transects were 6 cm greater than ALS median values. Moreover, the mean absolute difference and RMSD of that comparison was 7 and 8 cm, respectively. In sum, we will reference Currier et al. in the revision in the discussion on uncertainties in TLS-based snow depths.

Currier, W.R., Pflug, J., Mazzotti, G., Jonas, T., Deems, J.S., Bormann, K.J., Painter, T.H., Hiemstra, C.A., Gelvin, A., Uhlmann, Z., Spaete, L., Glenn, N.F., Lundquist, J.D., 2019. Comparing aerial lidar observations with terrestrial lidar and snow‐probe transects from NASA's 2017 SnowEx campaign. Water Resources Research. doi: 10.1029/2018WR024533.

4. There are a few comments in terms of structure of the paper. (1) Given the three research questions in L74-78, it would be best to either rearrange the results (discussion) to better address the three questions or increase the number of questions to better reflect the structure that the results are provided. Too many subsections exist in the result section. I would recommend combining the subsections based upon the questions. Also, please consider to combine "Results" and "Discussion" (because some descriptions in the both sections are duplicated). (2) I would strongly suggest reorganizing the figures and tables in the main body and the supplementary. I think some figures and tables (e.g. Figure 4 and Table 2) in the main body would be better to be

included in the supplementary. Similarly, some figures and tables in the supplementary should be moved into the main body (Figure B1). (3) Some sections should be re-named and relocated. For example, "2.1 Study area" should not be under "2.Method". "2.2 TLS" and "2.3 TLS Data Processing" should be combined which may be under "Data and preprocessing". Also, I think contents in some subsections are too short to comprise an individual subsection (e.g. Section 3.6 and 3.7). It would be good to combine similar subsections into one.

Thank you for the helpful comments, and this comment aligns with similar comments by Reviewer 1. We will review the questions and better present the results and discussion in a logical order. We may keep Results and Discussion separate, but will review what we present so to minimize duplication.

Regarding Figures and Tables - we will move Figure B1 to the manuscript, and move Figure 4 and Table 2 to the SI. Based on Review 1, we will also revise the caption of Figure B8 and move the information of Figure B9 into a table and we will merge the positive and negative violin plots of figure 7.

Regarding the sections, we will simplify the sections so that they flow better and avoid duplication.

Specific comments

L41 Please add citations We understand this comment is in regard to "and understanding how best to describe forest characteristics (cover, structure, gaps, etc.) relevant to snow distribution is evolving." and we will add citations.

L58 Please provide a range of the snow depth quantitatively (e.g. snow depth > XX mm) with general forest information (e.g. dominant types).

The dominant species was Pinus sylvestris and shallow snow here means snow depth < 0.5 m and deep snow is > 0.5 m. We will add this information to line 58.

L63-64 Would you check the reference again? Schirmer et al. (2011) do not provide

the relationship between vegetation or canopy characteristics and wind effect and snow depth variations.

Thanks. Yes, you are right. That was from Trujillo et al. 2007. We will modify the sentence.

L200 Can you quantify what "mid-to-high correlation" mean? Also I would recommend providing correlation matrixes between vegetation metrics and snow depth for each site to identify intercorrelation among the vegetation metrics.

Thanks. We will clarify what we mean by mid to high correlation. Our response will also be modified based on the changes we make from Comment #1 above.

L203 two distributions -> two peaks of the FHD distribution

Thanks for catching this mistake, we will correct for two peaks within the distribution.

L204 0.35-0.75 -> -0.35 to -0.75; throughout the manuscript

Thank you for catching this mistake. We will correct.

L222 Remove "we propose"

Will remove.

L304-205 This is identical to the sentence above. Rephrase "more evenly spaced along an individual tree"

Thank you for catching this, we will rephrase.

L273-274 Rephrase the sentence.

Yes, we will rephrase.

L292-294 This should be in data or methodology sections.

OK we will move it to the methodology section.

Table 1 In Median Absolute Deviation, what is the constant number "1.4826"?

1.4826 is a scale factor which relates MAD to deviation of average. This is MAD with one sigma uncertainty assuming normally distributed data. We will clarify in Table A1.

Table A2 Please add the units. I think boxplots would be more suitable to present the dataset. For example, 3x6 boxplots with three different colors for canopy, transition, and open areas.

Thanks for your suggestion. We will illustrate that table as a boxplot.

Figure 4 It would be fine to move into Appendix.

Yes we will move to Appendix.

Figure B9 In the figure, site O has random distribution. But in the caption "Only site N has a random distribution pattern of trees". Please double check. And I do not think the six distributions are needed in this figure even in supplementary info – all distributions are the same. A table including nearest neighbor values only would be more appropriate than the figure.

Thank you for finding this error. We will correct and change this figure into a table.

---

## Author Response (AR1)

Dear Reviewers,

Thank you for your helpful comments and which have resulted in a much improved manuscript. We've responded to the comments below in italics and would be happy to further clarify if necessary.

**RC1:**

Review of Hojatimalekshah et al., "Tree canopy and snow depth relationships at fine scales with terrestrial laser scanning"

Relating tree structure to snow depth dynamics is a very important research area with cryospheric and hydrologic implications. It is also a very challenging interaction to quantify and this study uses a Terrestrial Laser Scanning (TLS) dataset to examine such dynamics at a selection of site's from Grand Mesa in 2017. Interesting relationships are revealed in terms canopy and topographic controls. This paper would make a better contribution with a number of changes, generally around clarifying the storyline and focusing the results presented, and therefore I would recommend major revisions at this point prior.

My review begins with my major comments and followed by some specific comments.

At this point, I will not make technical comments.

**Literature context in Introduction:**

Overall, I enjoyed the introduction and how it established the context for the rest of the paper.

1.      This would further be strengthened by a concise description of the physical processes (interception/sublimation and blowing snow differences between forest and clearing) to help interpret the findings later.

*Thank you for the suggestion. We added details to the Introduction re: physical processes between forest and unforested areas.*

2.    As well, there is a focus on airborne lidar and terrestrial lidar as the only tools by which to quantify snow-forest interactions. There is a large chunk of recent literature that has been starting to employ drones (with structure from motion- many including Buhler 2016, Harder et al., 2016; Vander Jagt et al., 2015; De Michele et al., 2016, Walker et al., 2020. SfM does have trouble with dense vegetation but would still be a method in the sites with more sparse tree cover. More recently, lidar on drones for snow depth (Harder et al., 2020 and Jacobs et al., 2020) has been demonstrated to deal with vegetation better than ALS). Drone scales bridge TLS to airborne scales so is relevant in this discussion and should not be ignored. For full disclosure I do work with UAV's and snow and this is not reviewer coerced citation for selfish purposes– UAV scale and capabilities can and do work on these same scales/problems and should not be ignored.

*This is a good point and appreciate the references! We modified the Introduction and Discussion to include recent studies and demonstrate the value of UAS in snow-veg studies.*

**TLS for forests:**

1.    The path of photons from a TLS are oblique and so point densities quickly diminish as one moves further away the exposed side of a tree/tree stand and point densities diminish from the occlusion in vegetation. This is evident from the footprint of the analysis and the snow depth maps reported in the appendix. This is not discussed as a limitation anywhere that I can recall in this paper. TLS is not a new method to capture edge of forest snow interactions and I am not discounting previous work. I guess that I would like to see a discussion/acknowledgment of this. Especially when we are asked to consider the relationship with respect to side of tree. Won't this dataset be biased to capture interactions in greater detail on the exposure side rather than the obscured size. Will this bias the results/ relationships?

*We utilized multiple scan positions to avoid occlusion of the trees we used for analysis; however as you point out, there are some instances where occlusion occurred. We avoided this for our canopy edge analyses by only using tree polygons with sufficient coverage (see Section 3.1.4 line 175 and section 3.1.5, lines 206-207). We agree that it is important to add a statement about occlusion in our Discussion as a challenge of TLS, and we did this in the context of the complementary nature of UAS and airborne lidar (see lines 372-378).*

2.    How do the vegetation metrics respond to a TLS scan which captures one side of a tree better than the other? Also is there a threshold to determine the extent of the analysis and how is that determined?

*This is a good question and we'd like to clarify. We did not test for vegetation metrics based on a subsampling approach (on one side of a tree versus the other). In other words, to avoid unbalanced samples from different sides of a tree we used the whole tree to compute the related metrics. In addition, each tree was scanned from at least three different scan positions. However, there is still uncertainty about trees being evenly scanned specifically within dense canopies which we mention in the Discussion. Specific to our directional analysis, we used tree heights (only) to correlate to snow depth per each direction. While*

*tree heights could be affected by occlusion if scanned on only one side of the tree, we feel confident that our trees we used for analysis had the full tree height, based on our field sampling protocol, and knowledge and visual inspection of the data. We clarified these concerns in the Methods and Discussion.*

*Regarding the second question, we think you mean the extent (distance from scanner) to which the vegetation metrics can be calculated without significant error from sampling distances? We admit this is hard to judge given the variability at each site. The maximum distance between each pair of scan positions and any one scan position to the edge of the study area was less than 100 m; the only exception was site O where the distance between two of the scan positions in the middle of the site (open area) was 150 m. The beam divergence of our instrument is 0.3 mrad, equivalent to laser spot diameter of 3 cm at 100 m distance. Given the multiple scans and the vegetation cover, we feel comfortable suggesting that the extent of analysis for this environment is on average ~100 m from any one scan position.*

3.    Considering these ideas should this be paper/title be rather reframed as an analysis of snow depth-tree relationships for sparse/gappy areas. As it is presented it would seem that these findings should be pertinent to all forests but from the snow depth maps we can see that extent of the datasets are very much limited to isolated/sparse trees and edge of forest areas?

*We understand what you mean given that our images show plots that capture edges. Our analyses use individual trees in rather dense canopies (please see Table A2) and we utilize edges of canopies - we included this in the Discussion (see lines 371-372). We also mentioned in our Discussion that our interpretations cannot be used for general assumptions / forests outside this study area (See line 371). For these reasons, we prefer to retain the title as is.*

**Results:**

1.    There are a lot of results represented in various figures and tables that is on the overwhelming side of things. Can this be significantly pared down to the most important findings or the conversion of some large tables into figures (Table a1)? With 32 possible descriptor variables that have variable levels of description and no hypotheses presented it implies more of a fishing expedition which is not ideal. Are there metrics that have been used previously besides tree height? Could you focus on metrics that are a bit more common – ie LAI or sky view factor are some that come to mind- so that these findings could have broader applicability. FHD is not a commonly used metric yet. In a revision could this be flipped and some specific hypotheses be tested? Would help to focus things.

*Thank you – we followed your suggestion and replaced Table 2 with a more succinct table. We replaced Table A2 with Figure 6 and updated Tables 3-6 according to the change in our methods as the second reviewer requested. We added new figures (see Figures 4, 6, 7, 9, 10 and B1) in the main body and put appropriate information in them. The use of the different vegetation variables are in response to question 1 of the manuscript. However, we pared those down to three metrics (FHD, crown volume and cumulative vegetation return within the xth layer of the crown). FHD comes from the ecology community and while it may not be commonly used in cryosphere studies, we hope to introduce new approaches to this community. In our revision we provided more context for the use of FHD (Please see section 3.1.4).*

2.       Need more help with interpretation of Figure B8, B9 (text way too small), 7 (why are the positive and negative scales split).

*Thank you for the suggestion. We removed figure B8 as it was not informative to our results. We changed Figure B9 into Table A2 for clarity. We also removed Figure 7 as it contained repetitive information of Figures B3-B8 of the revised manuscript. In addition, we added Tables 5 and 6 which are easier to interpret the increasing and decreasing trends of snow depth from the edge of the canopy to the open.*

3.    Figure b1 should be more prominent – very useful for interpretation of all of this data.

*Thanks for the suggestion. We moved it to the manuscript (See Figure 5).*

4.       Could the results and discussion be grounded more strongly in the physical process descriptions – correlations themselves in specific situations can be hard to parse.

*Yes, we revised our process and used multiple linear regression and decision tree for more robust results and modified the Discussion to 'ground' our observations within the context of physical processes.*

**Specific comments:**

1.       I struggled to understand what was being communicated in the paragraph in lines 63-68. What does "observed the high contribution of storms in defining the snow accumulation pattern" mean in this context? Can it be clearer what proper scales are and how that is related to the controlling processes?

*Thanks. A previous study (Trujillo et al., 2007) has shown that wind and vegetation controls on snow depth are scale-dependent and wind may control snow depth distribution at larger scales than vegetation. However, these scales differ amongst other environmental parameters and within the range of meters to tens of meters. Overall we clarified the role of TLS and scale in lines 71-85*

*Trujillo, E., Ramírez, J. A. and Elder, K. J.: Topographic, meteorologic, and canopy controls on the scaling characteristics of the spatial distribution of snow depth fields, Water Resour. Res., 43(7), doi:10.1029/2006WR005317, 2007.*

2.       Line 120: Can you elaborate on "thus we reclassified these points manually using the software TerraScan". Not reproducible without knowing what the manual procedures implemented were.

*Manual classification included visually separating snow under the trees from tree trunks (see line 143). Yes, we agree this isn't completely reproducible but adding these details will help.*

3.    Section 2.3.2: perhaps a graphic could be used to explain M3C2?

*Yes, we did this (see Figure B1).*

4. Line 130-131: Can you elaborate on how transition zones were classified?

*Yes, we indicated that the transition zone consisted of defining a 10 m outer buffer beyond each tree polygon in the direction of the open (See lines 155-156).*

5. Line 155: "contain a minimum on 10 snow pixels." The tree polygons are variable in size yes? Perhaps this would be more robust if a percentage of area needed snow pixels? Is this how low density snow depth areas are removed from the analysis?

*Thank you for pointing this out. We replaced that with snow cover percent, which is the area of the tree polygon covered by snow in percent (See line 177). We chose a 50 % threshold and it removed several trees from our process but it did not change the results.*

6. Section 2.3.6: What tool/software was used?

*We used ArcMap (see line 193).*

7. Section 3.6: Did it result in deeper snow? You have the data to test this, correct?

*Yes, the mean snow depth is presented in Figure 6.*

8. Line 250: how representative are findings that are limited to one side of a dense stand of trees to make a comment on tree-snow directional dynamics. A limitation with TLS. Can findings be modified to account for the bias?

*Thanks for pointing this out - we need to clarify that our comments are specific to site A and refer the reader to Fig 8b, and reiterate the sampling limitation of TLS. Regarding accounting for the bias, we believe we have addressed this in the response to #5, above.*

**Site naming:**

1. Could the sites be named in a way that can simply convey some of their main features. Lettering doesn't convey much and would make tracking the relationships a bit easier.

*We agree this is onerous though we would prefer to keep the lettering as is, as they correspond to the SnowEx project naming convention.*

2. Observation temporal extent: Can it be emphasized more clearly that these were single measurements (not multi-temporal) and primarily reflect snow accumulations processes. Things will obviously be different if needing to account for snowmelt dynamics.

*Yes, thank you and we pointed this out more clearly in the methods and discussion (see lines 91 and 364-365).*

I appreciate the challenges in relating snow depth and vegetation metrics at fine scales with real world data and look forward to seeing your response.

*Thank you and we appreciate these helpful comments and references!*

**References:**

Bühler, Y., Adams, M. S., Bösch, R., and Stoffel, A.: Mapping snow depth in alpine terrain with unmanned aerial systems (UASs): potential and limitations, The Cryosphere, 10, 1075– 1088, https://doi.org/10.5194/tc-10-1075-2016, 2016

De Michele, C., Avanzi, F., Passoni, D., Barzaghi, R., Pinto, L., Dosso, P., Ghezzi, A., Gianatti, R., and Della Vedova, G.: Using a fixed-wing UAS to map snow depth distribution: an evaluation at peak accumulation, The Cryosphere, 10, 511–522, https://doi.org/10.5194/tc-10-511-2016, 2016.

Harder, P., Schirmer, M., Pomeroy, J., and Helgason, W.: Accuracy of snow depth estimation in mountain and prairie environments by an unmanned aerial vehicle, The Cryosphere, 10, 2559–2571, https://doi.org/10.5194/tc-10-2559-2016, 2016

Harder P., Pomeroy J.W. And Helgason W.D. (2020) Improving sub-canopy snow depth mapping with unmanned aerial vehicles: lidar versus structure-from-motion techniques. The Cryosphere: 14, pp. 1919-1935 DOI: 10.5194/tc-14-1919-2020

Jacobs, J. M., Hunsaker, A. G., Sullivan, F. B., Palace, M., Burakowski, E. A., Herrick, C., and Cho, E.: Shallow snow depth mapping with unmanned aerial systems lidar observations: A case study in Durham, New Hampshire, United States, The Cryosphere Discuss., https://doi.org/10.5194/tc-2020-37, in review, 2020.

Vander Jagt, B., Lucieer, A., Wallace, L., Turner, M., and Durand, D.: Snow Depth Retrieval with UAS Using Photogrammetric Techniques, Geosciences, 5, 264–285, https://doi.org/10.3390/geosciences5030264, 2015.

Walker B, Wilcox E, and Marsh P. ACCURACY ASSESSMENT OF LATE WINTER SNOW DEPTH MAPPING FOR TUNDRA ENVIRONMENTS USING STRUCTURE-FROM-MOTION PHOTOGRAMMETRY. Arctic Science. 0(ja): -.https://doi.org/10.1139/AS-2020-0006

**RC2:**

**General comments**

In this work, the authors focus on quantifying the roles of tree structure (and wind and topographic characteristics such as slope and aspect) in controlling snow depth variations using terrestrial laser scanning (TLS). The authors found that vegetation structural metrics (foliage height diversity) and wind are highly influential on spatial variability of snow depth. They also highlighted that windward slopes have greater impact on snow accumulation than vegetation features.

Overall, I enjoyed reading this manuscript. The authors identified interesting research questions and attempted to leverage the strength of the TLS data to fill in the scientific gaps. However, the current version of this work needs to be improved upon before publication in The Cryosphere is warranted. Major concerns are given below. I would recommend that this paper be returned for major revisions and specifically request inclusions of additional analysis with appropriate interpretations and reorganizing the structures of the manuscript for the Cryosphere community.

**Major comments:**

1. A major concern is that the methodology used to derive the conclusion in this study is entirely based on correlation coefficient in a linear regression without investigating inter-dependency among the physical variables. I do not think if the linear correlation based approach is enough to identify and to conclude the relationship between snow depth and the tree variables. For example, the snow depth with the distance from the canopy edge is not linear (see Figure 2 in Hardy Albert, 1995). A potential approach I would suggest can be regression tree or multivariable analysis (e.g. multiple linear/logistic regression analysis with the standardized coefficient) to quantify relative contribution of the vegetation metrics along with wind, topographic features (Molotch et al., 2005; Schneider et al., 2020). Also, it would be beneficial to provide variation in snow depth, key vegetation characteristics, and elevation, etc along representative transects for the sites. Please refer to Sturm Fig.2 in Sturm et al. (2001).

*Thank you, we appreciate the comments and references. In response to this comment and Reviewer 1 comments to reduce the number of vegetation metrics, we investigated the multicollinearity of the variables by using a variance inflation factor. We retained three vegetation metrics which describe the vegetation structure and had a variance inflation factor close to 1. We then used a multiple linear regression for sites A, F, K, M and N. We observed collinearity at site O between northness, eastness and elevation and used a decision tree to investigate the vegetation and topographical effects on snow depth under the canopy. In open areas, we applied a decision tree for all sites. You can see the representative transects for all sites as follow.*

*Regarding the transects, this is certainly feasible with the data, and from multiple directions. Below are examples. However, we chose not to include these given they would add another 6 figures and we already have an extensive number of figures in the manuscript and SI.*

[Figure]

[Figure]

[Figure]

[Figure]

·     Molotch, N. P., Colee, M. T., Bales, R. C. Dozier, J. 2005. Estimating the spatial distribution of snow water equivalent in an alpine basin using binary regression tree models: the impact of digital elevation data and independent variable selection. Hydrological Processes 19 (7), 1459–1479. doi:10. 1002/hyp.5586

·     Schneider, D., Molotch, N. P., Deems, J. S., Painter, T. H. (2020). Analysis of topographic controls on depletion curves derived from airborne lidar snow depth data.Hydrology Research.

·     Hardy, J. P., Albert, M. R. (1995). Snowâ˘ARˇ induced thermal variations around a single conifer tree. Hydrological processes, 9(8), 923-933.

·     Sturm, M., Holmgren, J., McFadden, J. P., Liston, G. E., Chapin III, F. S., Racine, C. H. (2001). Snow–shrub interactions in Arctic tundra: a hypothesis with climatic implications. Journal of Climate, 14(3), 336-344.

2.   Regarding the comment above, another concern is the interpretation of the correlation coefficient values from the linear regressions in the result sections. For example, the authors state that "slope explained 44

*While this comment was cut off, we think the reviewer is concerned about 'slope explained 44% of the variance' and how this is interpreted. This is clarified in the process of changing our methods to use the coefficients and R-squared of multiple linear regression.*

3.   I think the TLS data's reliability should be verified. How accuracy is TLS-based snow depth, especially under the tree canopy? Many previous studies found that there were issues in ALS, TLS, structure-from-motion photogrammetry (SfM) with observation gaps in forested regions. The return density under or near canopy can be extremely low that may not be adequate to observe spatial variations of snow depth. In Figure B1, the snow depth maps the authors provided seem to be very limited to areas near forest and under the trees. Thus, I would recommend that the author quantify the accuracy of the snow depth measurements especially under/near canopy. Have you seen comparison results with independent snow depth measurements? I know the validation work is out of scope in this study, but it would be helpful for readers to be able to have a sense of how accuracy the TLS technique is, particularly in these sites. I believe there are various available ground-based (or other techniques) snow depth measurements because this work was part of the NASA-led SnowEx 2017 campaign. If they are not available, the authors should provide at least general uncertainties in TLS-based snow depth from previous findings, particularly under/near canopy.

*Thank you for the comment. A previous paper (Currier et al., 2019) demonstrates the relative accuracy of the TLS to airborne lidar (ALS), which is likely a better measure than using the snow depth measurements that were not collected at all sites and/or same day. Currier et al., indicate that the median snow depth difference between ALS and TLS at sites A and K were less than 5 cm. In addition, ALS snow depth comparisons with field transects indicated that the median values for transects were 6 cm greater than ALS median values. Moreover, the mean absolute difference and RMSD of that comparison were 7 and 8 cm,*

*respectively. We referenced Currier et al., 2018 in the revision in the method on uncertainties in TLS-based snow depths (see lines 132-134).*

*Currier, W.R., Pflug, J., Mazzotti, G., Jonas, T., Deems, J.S., Bormann, K.J., Painter, T.H., Hiemstra, C.A., Gelvin, A., Uhlmann, Z., Spaete, L., Glenn, N.F., Lundquist, J.D., 2019. Comparing aerial lidar observations with terrestrial lidar and snow-probe transects from NASA's 2017 SnowEx campaign. Water Resources Research. doi: 10.1029/2018WR024533.*

4. There are a few comments in terms of structure of the paper. (1) Given the three research questions in L74-78, it would be best to either rearrange the results (discussion) to better address the three questions or increase the number of questions to better reflect the structure that the results are provided. Too many subsections exist in the result section. I would recommend combining the subsections based upon the questions. Also, please consider to combine "Results" and "Discussion" (because some descriptions in the both sections are duplicated). (2) I would strongly suggest reorganizing the figures and tables in the main body and the supplementary. I think some figures and tables (e.g. Figure 4 and Table 2) in the main body would be better to be included in the supplementary. Similarly, some figures and tables in the supplementary should be moved into the main body (Figure B1). (3) Some sections should be renamed and relocated. For example, "2.1 Study area" should not be under "2.Method". "2.2 TLS" and "2.3 TLS Data Processing" should be combined which may be under "Data and preprocessing". Also, I think contents in some subsections are too short to comprise an individual subsection (e.g. Section 3.6 and 3.7). It would be good to combine similar subsections into one.

*Thank you for the helpful comments, and this comment aligns with similar comments by Reviewer 1.*

*We reordered the results according to the questions. We kept Results and Discussion separate, but reviewed what we presented to minimize duplication. We also merged and simplified the sections so that they flow better.*

*Regarding Figures and Tables - we moved Figure B1 to the manuscript (see figure 5), and moved Figure 4 and Table 2 to the SI. We removed Figures 7 and B8 and moved the information of Figure B9 into Table A2.*

**Specific comments**

L41 Please add citations

*We understand this comment is in regard to "and understanding how best to describe forest characteristics (cover, structure, gaps, etc.) relevant to snow distribution is evolving." and we added citations.*

L58 Please provide a range of the snow depth quantitatively (e.g. snow depth > XX mm) with general forest information (e.g. dominant types).

*The dominant species was Pinus sylvestris and shallow snow here means snow depth < 0.5 m and deep snow is > 0.5 m. We added this information to lines 79-81.*

L63-64 Would you check the reference again? Schirmer et al. (2011) do not provide the relationship between vegetation or canopy characteristics and wind effect and snow depth variations.

*Thanks. Yes, you are right. That was from Trujillo et al. 2007. We modified the sentence (see lines 59-61).*

L200 Can you quantify what "mid-to-high correlation" mean? Also I would recommend providing correlation matrixes between vegetation metrics and snow depth for each site to identify intercorrelation among the vegetation metrics.

*Thanks. We replaced that by multiple linear regression results and intercorrelation among the metrics is evaluated by a variance inflation factor (metrics are independent if the variance inflation factor is 1 or close to 1).*

L203 two distributions -> two peaks of the FHD distribution

*Thanks for catching this mistake, we corrected for two peaks within the distribution.*

L204 0.35-0.75 -> -0.35 to -0.75; throughout the manuscript

*Thank you for catching this mistake. We replaced that with our multiple linear regression results.*

L222 Remove "we propose"

*Removed.*

L304-205 This is identical to the sentence above. Rephrase "more evenly spaced along an individual tree"

*Thank you for catching this, we rephrased.*

L273-274 Rephrase the sentence.

*Yes, we rephrased.*

L292-294 This should be in data or methodology sections.

*OK we moved it to the methodology section (see lines 238-240).*

Table 1 In Median Absolute Deviation, what is the constant number "1.4826"?

*We removed this variable, as we do not use it in our revised methodology. Regardless, to answer the question: 1.4826 is a scale factor which relates MAD to deviation of average. This is MAD with one sigma uncertainty assuming normally distributed data.*

Table A2 Please add the units. I think boxplots would be more suitable to present the dataset. For example, 3x6 boxplots with three different colors for canopy, transition, and open areas.

*Thanks for your suggestion.. We illustrated the table as a distribution plot because the variance and snow depth difference between three zones was clearer than in a boxplot.*

Figure 4 It would be fine to move into Appendix.

*Yes we moved to Appendix.*

Figure B9 In the figure, site O has random distribution. But in the caption "Only site N has a random distribution pattern of trees". Please double check. And I do not think the six distributions are needed in this figure even in supplementary info – all distributions are the same. A table including nearest neighbor values only would be more appropriate than the figure.

*Thank you for finding our error. We changed this figure into Table A2.*